# Tunable $CO_2$ electroreduction to ethanol and ethylene with controllable interfacial wettability

Yan Lin[1,2,3,5], Tuo Wang [1,2,3,4,5], Lili Zhang[1,2,3], Gong Zhang [1,2,3], Lulu Li[1,2,3], Qingfeng Chang[1,2,3], Zifan Pang[1,2,3], Hui Gao[1,2,3], Kai Huang[1,2,4], Peng Zhang [1,2,3,4], Zhi-Jian Zhao [1,2,3], Chunlei Pei[1,2,3] & Jinlong Gong [1,2,3] ✉

The mechanism of how interfacial wettability impacts the $CO_2$ electroreduction pathways to ethylene and ethanol remains unclear. This paper describes the design and realization of controllable equilibrium of kinetic-controlled *CO and *H via modifying alkanethiols with different alkyl chain lengths to reveal its contribution to ethylene and ethanol pathways. Characterization and simulation reveal that the mass transport of $CO_2$ and $H_2O$ is related with interfacial wettability, which may result in the variation of kinetic-controlled *CO and *H ratio, which affects ethylene and ethanol pathways. Through modulating the hydrophilic interface to superhydrophobic interface, the reaction limitation shifts from insufficient supply of kinetic-controlled *CO to that of *H. The ethanol to ethylene ratio can be continuously tailored in a wide range from 0.9 to 1.92, with remarkable Faradaic efficiencies toward ethanol and multi-carbon ($C_{2+}$) products up to 53.7% and 86.1%, respectively. A $C_{2+}$ Faradaic efficiency of 80.3% can be achieved with a high $C_{2+}$ partial current density of 321 mA cm$^{-2}$, which is among the highest selectivity at such current densities.

The electrochemical conversion of carbon dioxide ($CO_2$) into value-added multi-carbon ($C_{2+}$) products is an attractive route to close the carbon loop[1,2]. However, the formation of $C_{2+}$ products with high selectivity is still meeting great challenges[3]. *CO (* denotes the adsorbed species on the surface) and *H are considered to be the key intermediates during producing $C_{2+}$ products[4]. Previously, *CO coverage has been widely accepted as one crucial factor in the selective produce ethylene[5,6]. While, as another important $C_{2+}$ products, tuning the *H coverage is an effective approach to realize high-efficiency $CO_2$-to-ethanol conversion[7,8], which is less relevance with *CO coverage[5]. Therefore, the role of competitive *CO and *H intermediates on the catalyst surface on pathways of ethylene and ethanol needs further understanding.

As the only metal with a negative adsorption energy for *CO but a positive adsorption energy for *H, copper (Cu) presents a unique ability to produce $C_{2+}$ products[4,9]. However, multiple products have been detected on Cu surfaces resulting in poor product selectivity for Cu[9,10]. Previous investigations have found that the coverage of *CO and *H on the Cu surface plays a critical role in determining the selectivity of $C_{2+}$ products[4,11]. Enormous effort has been devoted to catalyst design for selective ethylene or ethanol production, including facet control[12–14], surface morphology tuning[15–17], bimetallic and multi-metallic alloying[8,18,19], metal/metal compound hybrids construction[7], which can be employed to alter the adsorption or coverage of key intermediates. Generally, the facet control and surface morphology are only capable to modulate the *CO adsorption[13,15,16]. Moreover, the introduction of CO- or $H_2$-producing metals in catalysts can increase

[1]School of Chemical Engineering & Technology, Key Laboratory for Green Chemical Technology of Ministry of Education, Tianjin University, 300072 Tianjin, China. [2]Collaborative Innovation Center for Chemical Science & Engineering (Tianjin), 300072 Tianjin, China. [3]Haihe Laboratory of Sustainable Chemical Transformations, 300192 Tianjin, China. [4]Joint School of National University of Singapore and Tianjin University, International Campus of Tianjin University, 350207 Binhai New City, Fuzhou, China. [5]These authors contributed equally: Yan Lin, Tuo Wang. ✉e-mail: jlgong@tju.edu.cn

the coverage of *CO or *H, leading to increased by-products of $H_2$ and $CO$[8,19]. Furthermore, these strategies may increase the hydrophilicity of the electrode, resulting in excessively high *H coverage[7]. Therefore, it remains a great challenge to simultaneously modulate the coverage of *CO and *H with these strategies.

Moreover, local $CO_2/H_2O$ concentration derived from the mass transport of $CO_2$ and $H_2O$ can influence the surface coverage of *$CO_2$ (a precursor of *CO), *CO, and *H, which affects the reaction pathways toward $C_{2+}$ products[5,20]. The superior $CO_2$ mass transport in gas diffusion electrode (GDE)-based systems can be achieved due to the shorter diffusional lengths than that in conventional H-type cells[5]. However, the catalysts in GDE are covered by a thin layer of electrolyte due to their metallic nature that leads to an intrinsic hydrophilic surface, causing the reaction primarily occurs with dissolved $CO_2$ in the aqueous phase[21–24]. Thus, gas diffusion in the catalyst layer becomes the limiting step of the cathode mass transport[20,25]. The poor solubility of $CO_2$ in water leads to the rapid exhausting of $CO_2$ in the thin electrolyte layer, which changes the local $CO_2/H_2O$ supply and limited $C_{2+}$ products. To break the mass transport limitation and improve the selectivity of $C_{2+}$ products, GDE can be modified to improve the transport of reactants and control the coverage of key intermediates[20,26–32]. Organic molecules (fluorosilane[33], quaternary ammonium salt[29,30]), polymers (polytetrafluoroethylene[20,28]), and ionic polymers (Nafion[26,31]) can modulate the local concentration of $CO_2$ and $H_2O$ because their backbone chains containing $-CH_2-$ or $-CF_2-$ induce hydrophobicity. Although these long-side chains can induce strong hydrophobicity on catalyst surfaces to improve mass transport effectively, continuous control of wettability is still meeting great challenges[20,28,33]. Moreover, although the $C_{2+}$ selectivity can be enhanced by using hydrophobic polytetrafluoroethylene (PTFE) layers, it is difficult to identify and optimize the specific effect of hydrophobicity[27]. Thus, the impact of interfacial wettability on the pathways of products, especially ethylene, and ethanol, is rarely understood. Therefore, it is urgently desirable to develop a new approach to continuously tune the kinetic-controlled *CO/*H ratio by altering the interfacial wettability.

This work describes the design and realization of tunable interfacial wettability through using alkanethiols with different alkyl chain lengths. Then, the local concentration of $CO_2$ and $H_2O$ can be modulated by changing $CO_2$ and $H_2O$ transport through different interface wettability, which may achieve an optimized equilibrium of kinetic-controlled *CO and *H in a controllable manner. Thiol can be anchored on the Cu surface through coordination binding[34]. Wettability can be controlled by the length of an alkyl chain, which can effectively block the absorption of water and facile $CO_2$ transport. As a result, the ratio of produced ethanol to ethylene can be tuned in the wide range after hydrophobic treatment (from 0.90 to 1.92) with remarkable Faradaic efficiencies (FE) of ethanol and $C_{2+}$ (up to 53.7% and 86.1%, respectively). Based on the establishment of this structure, the effect of interfacial wettability on the pathways of ethylene and ethanol is further revealed through in-situ spectroscopy and computational fluid dynamic simulation.

## Results

### Characterization of the wettability-tunable modified Cu electrode

The copper catalysts were prepared by direct current (DC) sputtering with a metallic copper target in Ar on carbon paper as the substrate (details in the "Methods" section). To fabricate a modified layer with continuously adjustable wettability, the Cu surface was modified by alkanethiols with different lengths of an alkyl chain, which were denoted as Cu-4C, Cu-7C, Cu-12C, and Cu-18C (Cu-xC, x stands the number of carbon (C) atoms in the alkyl chain, details in the "Methods" section). As shown in scanning electron microscopy (SEM) images, Cu nano-islands are uniformly grown on the carbon paper surface with an

average size of around 1 μm (Supplementary Fig. 2). The cross-sectional SEM image also shows that the thickness of the Cu layer is around 1 μm (Supplementary Fig. 3). X-ray diffraction (XRD) indicates that both Cu catalysts tend to expose Cu(111) facet (Fig. 1a). TEM images and elemental energy-dispersive x-ray spectroscopy mapping (EDX) also reveal a 2–3 nm continuous and uniform alkanethiol layer with wettability modification (Fig. 1b). The modification of alkanethiol on Cu is further confirmed through attenuated total reflection surface-enhanced infrared absorption spectroscopy (ATR-SEIRAS) (Supplementary Fig. 4). The Cu−S coordinate bonds were formed via alkanethiolation as illustrated by X-ray photoelectron spectroscopy (XPS) of Cu 2p and Auger electron spectroscopy (AES) of Cu LMM. Before modification, the Cu catalyst consists of $Cu^+/Cu^0$ (932.5/952.4 eV) and $Cu^{2+}$ (934.8/954.6 and 943.6/962.7 eV) (Fig. 1d, Supplementary Fig. 5)[35]. For a better comparison, the surface of the bare Cu sample was analyzed in XPS with Ar etching. After Ar etching, the bare Cu sample presents Cu 2p peaks at 932.5 and 952.4 eV, corresponding to $Cu^+/Cu^0$ species (Supplementary Fig. 6a). The AES of Cu LMM further indicates that the bare Cu sample mainly consists of $Cu^0$ (918.6 eV) after Ar etching (Supplementary Fig. 6b)[36,37]. After modification, the $Cu^{2+}$ species are replaced by $Cu^+$ species (932.5/952.4 eV), and a new S 2p peak is observed at 163.0 eV, which can ascribe to the generation of Cu−S bonds (Fig. 1e)[35,38]. Furthermore, the relevance between the alkyl chain length of alkanethiol and the contact angle was established. The contact angle of alkanethiol-treated Cu can be controllably increased by varying the length of the alkyl chain, and the contact angles are arranged at 43°, 95°, 112°, 131°, and 156°, respectively (Fig. 1f, Supplementary Table 1). Thus, a modified layer with continuously tunable wettability, as well as a homogeneous structure, was obtained.

### Effect of controllable wettability on $CO_2$ electroreduction performance

To assess the impact of the wettability on product distribution, the electrochemical performance of Cu catalysts with different wettability was evaluated in a three-compartment flow cell (Supplementary Fig. 8). FE of $H_2$ evolution on hydrophobic-treated Cu catalyst decreases rapidly to below 10% (Fig. 2a). Simultaneously, the FE for CO drops from 26.1% to 2.7% and then increases to 8.9%. With the increasing of hydrophobicity (contact angle: from 43° to 131°), the FE of $C_{2+}$ (ethylene, ethanol, propanol) increases from 55.4% to 86.1%, with corresponding current density from 91.4 to 103.3 mA cm$^{-2}$ (Fig. 2b). Interestingly, as the hydrophobicity further increases to a contact angle of 156°, the FE and partial current density of $C_{2+}$ decrease significantly. With the increasing current density, the $C_{2+}$ Faraday efficiency remains at a high level, leading to an increased partial current density of $C_{2+}$ products. Thus, a $C_{2+}$ Faradaic efficiency of up to 80.3% can be maintained even at a high current density of 400 mA cm$^{-2}$, corresponding to a $C_{2+}$ partial current density of 321 mA cm$^{-2}$, which is among the best performances (Fig. 2c, Supplementary Table 2). Furthermore, the ethanol-to-ethylene ratio increases from 0.90 to 1.93 and subsequently drops to 1.13 with the increasing of contact angle from 43° to 131°. The FE of ethanol is enhanced dramatically, from 23.8% to 53.7%, exceeding most reported Cu-based electro-catalysts (Fig. 2d, e, Supplementary Table 2). Under super-hydrophobicity (CA = 156°), Cu catalyst exhibits a notable increase of FE toward ethylene, whereas production of ethanol is suppressed obviously (Fig. 2e). Moreover, the variation trend of methane Faradaic efficiency is consistent with that of ethylene, while the variation trend of propanol Faradaic efficiency is similar as that of $C_2$ products (Supplementary Fig. 10). Here, the mechanism of wettability on modulating ethanol and ethylene will be discussed below.

To confirm the relevance of selectivity with wettability, the accompanied chemical structure and morphology variations should be analyzed and excluded. The durability of the alkanethiol monolayer

and Cu electrode was evaluated by XRD, XPS, TEM, contact angle measurement, and OH⁻ electroadsorption measurement. The 6.5 h stability test at −1.2 V versus (vs.) reversible hydrogen electrode (RHE) shows that the stable reduction current density and product selectivity are maintained after alkanethiol modification (Supplementary Fig. 17). According to the SEM images, XRD profiles and OH⁻ electroadsorption measurement, the morphology and dominant exposed facets of Cu catalyst are not significantly changed (Supplementary Figs. 18–20). XPS analysis before and after the electrolysis reveal similar Cu/S ratios on the catalyst surface (Supplementary Fig. 21). The HRTEM displays the presence of the alkanethiol layer on the Cu catalyst after the reaction due to the high electrochemical stability of the hydrophobic layer (Supplementary Fig. 22). The hydrophobic-treated electrode can maintain a larger contact angle after electrolysis, which ascribes to the optimized wettability of the catalyst layer that improves the $CO_2RR$ stability (Supplementary Fig. 23). In this regard, the exposed facet, morphology and hydrophobicity of the Cu catalyst and the thiol layer after electrolysis are relatively stable under experimental condition, which is in accordance with the previous report[27,34]. Additionally, the electrochemically active surface area (ECSA) also drops rapidly after hydrophobic treatment (Supplementary Fig. 24), although the Brunauer−Emmett−Teller specific surface area ($S_{BET}$) is almost unchanged (Supplementary Fig. 26), consistent with previous result[27]. Therefore, the lower ECSA and current density with the formation of the alkanethiol layer can be attributed to the reduced contact with the aqueous electrolyte (Fig. 2f, Supplementary Fig. 24, 25). The variation of charge resistance ($R_{ct}$), mass transport resistance ($W$), and interfacial electrochemical double-layer capacitance ($C$) after alkanethiol modification in electrochemical impedance spectroscopy (EIS)

analysis can be ascribed to the strong hydrophobicity that retarded the contact with the electrolyte after alkanethiol-modification, which is confirmed by the ECSA analysis. (Supplementary Fig. 27, Supplementary Table 4). More importantly, formate is the main product of the S-Cu-composed electrode in previous studies[39–41]. However, the catalyst in this study contributes to the improved $C_{2+}$ product, which is ascribed to the new reduction pathways of $CO_2RR$ after wettability modification. Thus, the chemical structure or morphology changing after thiol-modified is not the reason for improved $CO_2RR$ selectivity of ethylene and ethanol.

### Effect of controllable wettability on *CO coverage via $CO_2$ mass transport

It is known that the mass transport of $CO_2$ (local $CO_2$ concentration) can affect the coverage of *$CO_2$ (a precursor of *CO), *CO, and *H, which affects the reaction pathways toward ethylene and ethanol further[5,20,42]. Our data (Fig. 2) also may imply that the kinetic-controlled *CO/*H ratio can be controlled by tuning the local $CO_2/H_2O$ ratio through changing the wettability of the catalyst. To circumvent this issue, computational fluid dynamic (CFD) simulations were employed to investigate the mass transport of $CO_2$ and $H_2O$ in the catalyst layers related to kinetic-controlled *CO and *H. The models with and without the alkanethiol modification layer were established to quantity $CO_2$ along the catalyst surface (Fig. 3a). The difference in gas diffusion was verified by simulating $CO_2$ concentration. The availability of the gas reactant significantly varied at the gas−electrolyte interface after modification. Comparing with the unmodified-hydrophilic Cu catalyst, the gas diffusion distance of the alkanethiol-modified layer is increased due to the improved gas diffusion.

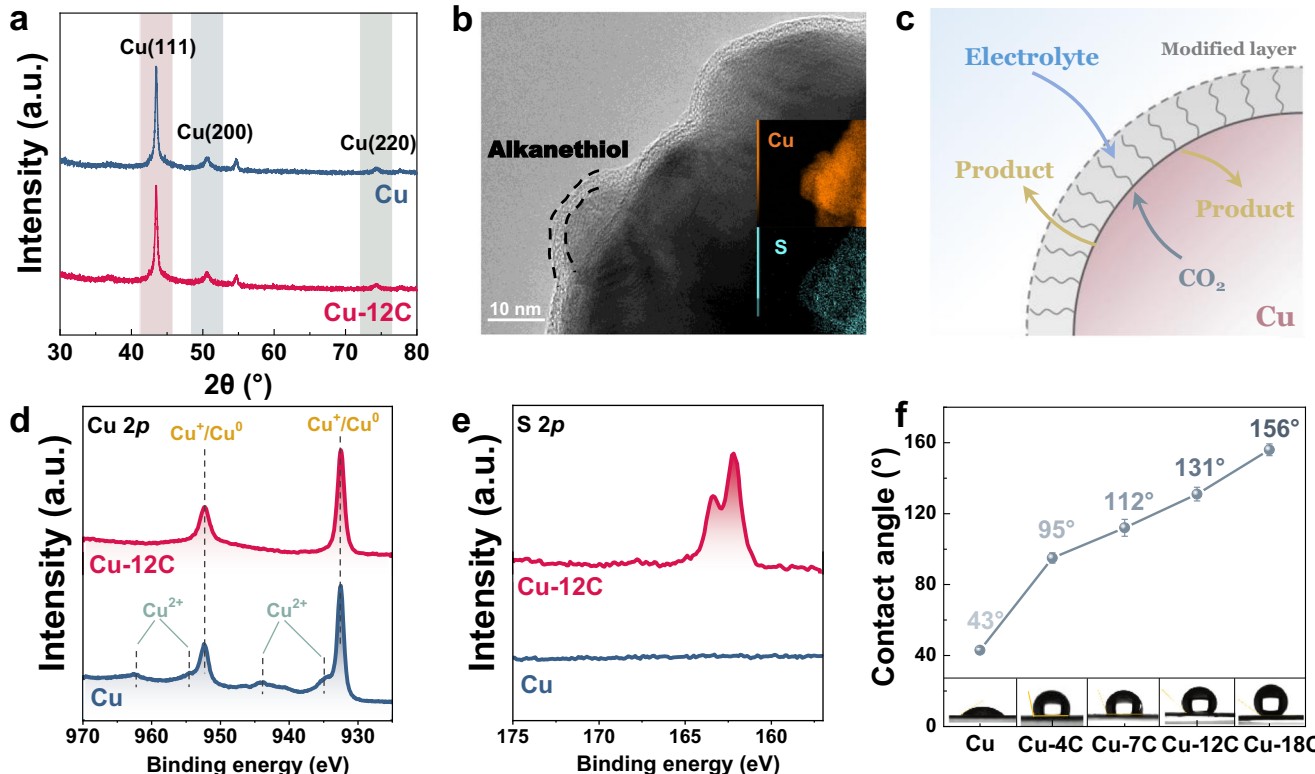

**Fig. 1 | Characterization of the wettability-tunable modified Cu electrode.** **a** XRD patterns of Cu and Cu-12C show that both Cu catalysts with or without modification tend to expose Cu(111) facet. **b** HRTEM image of Cu-12C, revealing a 2–3 nm continuous and conformal alkanethiol layer. **c** Illustration shows Cu catalysts via hydrophobic treatment by alkanethiol. **d** Cu 2*p* XPS spectra. **e** S 2*p* XPS spectra of Cu and Cu-12C, demonstrating the formation of Cu−S bonds. **f** The average water droplet contact angle of alkanethiol-treated Cu increases with the increase of the alkyl chain length. The insets show photographs of water droplets. A modified layer with continuously tunable wettability, as well as a homogeneous structure, was obtained. The a.u. stands for arbitrary units. Error bars represent the standard deviation from at least three independent measurements.

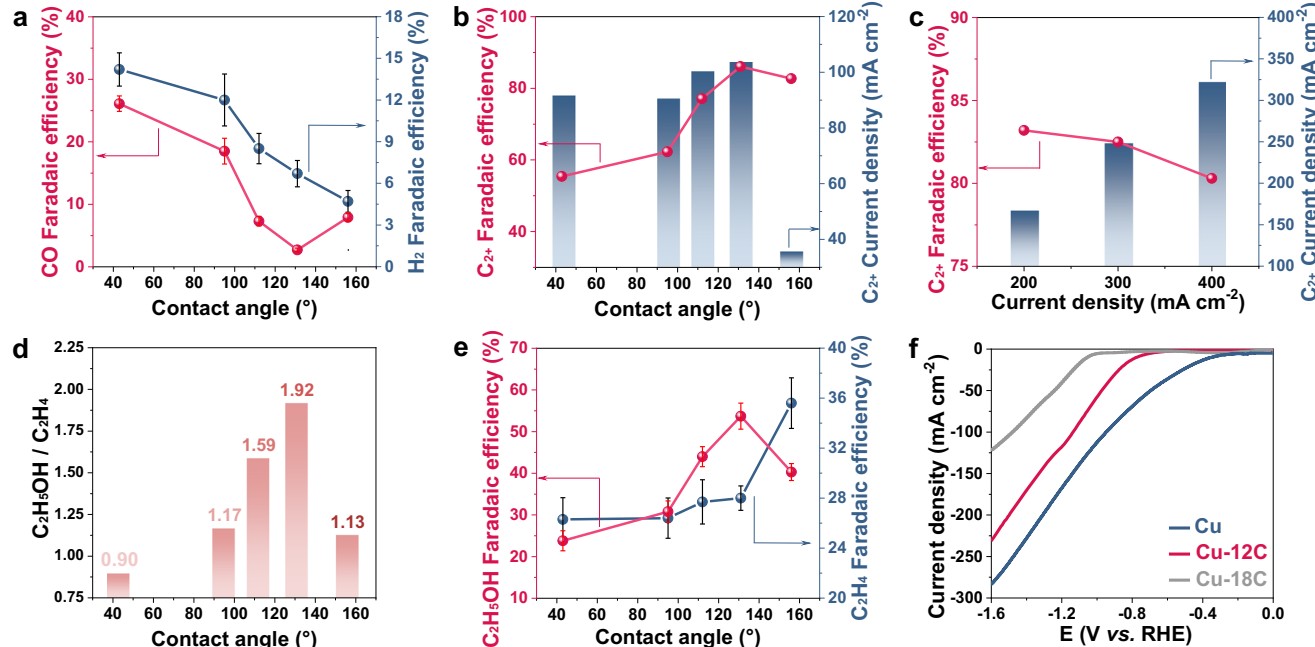

**Fig. 2 | Effect of controllable wettability on $CO_2$ electroreduction performance.** **a** Faradaic efficiencies of CO and $H_2$. **b** Faradaic efficiencies and partial current densities of $C_{2+}$ on Cu electrodes in 1 M KOH (aq.) at −1.2 V versus (vs.) reversible hydrogen electrode (RHE) with various contact angles **c** Faradaic efficiencies and partial current densities of $C_{2+}$ on Cu-12C under different current densities. **d** Ethanol to ethylene ratios and **e** Faradaic efficiencies of ethanol and ethylene on Cu electrodes in 1 M KOH (aq.) at −1.2 V vs. RHE with various contact angles (without iR correction). It is demonstrated that the strong dependence of the $C_2$ product distribution on the wettability of Cu catalyst. The evolution of $H_2$ and CO is suppressed via alkanethiol treatment. **f** Linear sweep voltametric (LSV) curves of the Cu electrodes with and without alkanethiol modification in 1 M KOH ($v$ = 50 mV s$^{-1}$), showing the lowered current of the hydrophobic-treated Cu catalyst at a given potential. Error bars represent the standard deviation from at least three independent measurements.

Accordingly, the effect of $CO_2$ mass transport on local $CO_2$ concentration needs to be further studied. However, it is difficult to measure local $CO_2$ concentration in the catalyst layer directly. To investigate the local $CO_2$ concentration of different interfacial wettability, control samples were prepared with different gas–liquid–solid contact, enabling in situ measurements with fluorescence electrochemical spectroscopy (FES) at 100 mA cm$^{-2}$ chronopotentiometry (Fig. 3b). The local $CO_2$ concentration of all samples decreases to a new steady state during electrolysis. As for the three-phase contact electrode, increasing hydrophobicity can shorten the relaxation time of the local $CO_2$ concentration and increase equilibrium concentration, which can facile $CO_2$ mass transport. The local $CO_2$ concentration will return back to the initial equilibrium value after electroreduction (Supplementary Fig. 30). The recovery of local $CO_2$ concentration on the highly hydrophobic interface is faster due to the improved gas diffusion. These results demonstrate that the interfacial contact state affects the $CO_2$ mass transport from the bulk phase to the catalyst, thus affecting the local $CO_2$ concentration nearby the catalyst.

The hydrophobic treatment also can promote CO diffusion in CO reduction reactions. Thus, it can be excluded that the limited $CO_2$ mass transport at the hydrophilic Cu electrode is entirely due to the neutralization of $CO_2$ with the electrolyte (Fig. 3c, Supplementary Fig. 31). Like in the $CO_2$RR, the direct coupling between two *CO species in the CORR is widely accepted as the major C−C coupling mechanism. Moreover, *CO coverage is determined by the local concentration of CO near the catalyst. Thus, the coverage of key intermediate *CO can be improved by directly using CO as reactant. When the Cu catalyst has a similar contact angle, the promotion of ethanol production is more obvious than that of ethylene during CORR. In comparison, a similar phenomenon can be observed on CuAg catalyst, which can generate more CO during performing $CO_2$RR (Supplementary Fig. 32). Therefore, increasing *CO coverage will benefit ethanol generation.

The local $CO_2$ concentration can further influence the coverage of *CO, which affects the reaction pathways toward ethylene and ethanol. Thus, in-situ ATR-SEIRAS was employed to further evaluate the impact of interfacial wettability on the adsorption of intermediates (Fig. 3d, e). There is a stretching band at ~2070 cm$^{-1}$ on all electrodes, corresponding to the stretching band of the linear-bond CO ($CO_L$) on the Cu surface[43,44]. As for hydrophobic-treated Cu electrodes, the $CO_L$ bands become more clearly defined at moderate overpotentials (−0.6 to 1.2 V vs. RHE), indicating a larger *CO coverage. Furthermore, $CO_L$ bands are preferentially observed on hydrophobic-treated Cu surfaces under more negative potentials, with a wider voltage range of *CO coverage[14,45]. The enhanced *CO adsorption can be attributed to the high production rate of the accumulation of more *CO on a hydrophobic Cu surface under a high local concentration of $CO_2$[31]. The difference in *CO coverage over a wide voltage range derived from the wettability of the catalyst layer is one of the key factors for the ethylene and ethanol pathways.

## Effect of controllable wettability on *H coverage via $H_2O$ mass transport

As another key intermediate, the *H intermediate on the surface is converted from the bulk solution, which is affected by $H_2O$ transport. In order to elucidate the effect of $H_2O$ transport on the ethylene and ethanol reaction pathways, the hydrogen evolution reaction (HER) performance of Cu catalysts with different wettability for various electrolytes was compared (Fig. 4a, b). The HER performance should not be affected by the external gas diffusion but is determined by $H_2O$ availability and transport[26]. Therefore, stronger hydrophilicity can benefit the HER due to more efficient $H_2O$ transport.

Unfortunately, the peak of *H on Cu cannot be observed by in-situ Raman spectra (Supplementary Fig. 35) or in-situ ATR-SEIRAS spectra (Fig. 3d, e) due to the extremely weak adsorption of *H on Cu. At present, *H coverage can be investigated indirectly through *CO

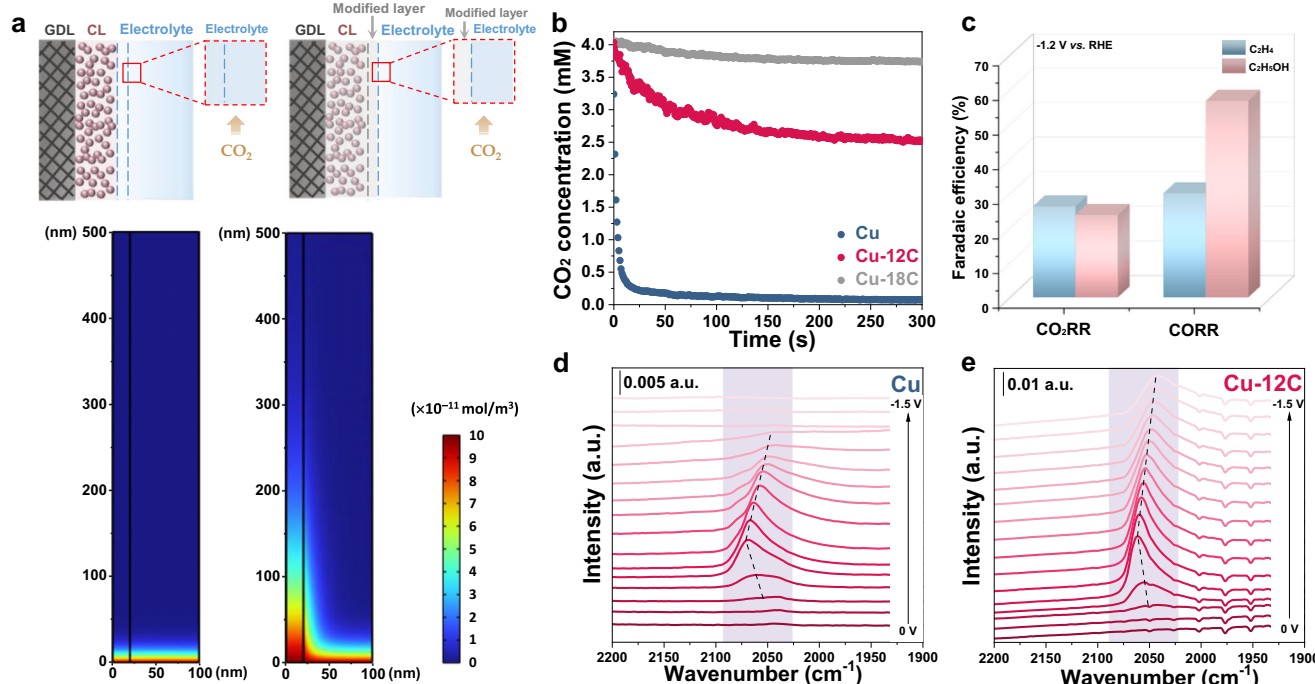

**Fig. 3 | Effect of controllable wettability on \*CO coverage via CO₂ mass transport. a** Top row: schematics display the CO₂RR configuration with (right column) or without (left column) alkanethiol modification. The red dotted box is the simulation area. Second row: comparison of modeled gas availability along the catalyst surface with (right column) or without (left column) alkanethiol modification. Gas availability dramatically increases via an alkanethiol-derived hydrophobic environment. GDL: gas diffusion layer, CL: catalyst layer. **b** Local CO₂ concentration versus time during the CO₂RR of 100 mA cm⁻² of modeled Cu, Cu-12C and Cu-18C interfacial environments, respectively. A stronger hydrophobicity indicates a faster CO₂ mass transport. **c** Comparison of ethylene and ethanol Faradaic efficiencies for CO₂RR versus CORR on Cu, which indicates the improvement of \*CO coverage is more favorable for the ethanol pathway than for ethylene. In-situ ATR-SEIRAS spectra of **d** Cu and **e** Cu-12C, revealing that higher \*CO coverage can be maintained on Cu electrodes after hydrophobic treatment. Wherein the stretching band at ~2070 cm⁻¹ corresponds to the stretching band of CO_L on Cu surface. The a.u. stands for arbitrary units.

coverage or H₂O content[18,46–49]. \*H and \*CO occupy most of the Cu surface sites, resulting in direct competition between \*H and \*CO for surface sites[47]. When the surface maintains a high \*CO coverage, the corresponding \*H coverage will decrease[12,18,46,47]. According to the results of in-situ ATR-SEIRAS spectra (Fig. 3d, e) and in-situ Raman spectra (Supplementary Fig. 35), higher hydrophobicity indicates higher \*CO coverage, thus the corresponding \*H coverage is lower. Further, the surface coverage of intermediate (\*CO, \*H) is directly proportional to the concentration of reactants (CO₂, H₂O)[5,42]. Therefore, the \*H coverage can be deduced indirectly through water content. The confocal laser scanning microscopy measurements (CLSM) were employed to observe the variation of available water near the interface (details in the "Methods" section) (Fig. 4d–f)[48]. To determine the relevance of absorbed water with different wettability, cross-sectional images along the z-axis are chosen for comparison. The green fluorescent regions represent the electrolyte phase, while the dark regions represent the gas phase or solid electrode. The decay tailing effect of fluorescence intensity at the liquid phase boundary represents the decreasing of available water (since the H during CO₂RR source from H₂O, which can represent \*H coverage), which can be valued by decay distance. The linear sweep plots of fluorescence intensity along the z-axis at the white arrows from the cross-sectional images were obtained. Without modification, the electrolyte can penetrate into the hydrophilic surface of the Cu catalyst. The liquid–solid interface occupy the whole surface of the catalyst (Cu) rapidly due to the hydrophilic Cu surface. On the contrary, the catalyst layer (Cu-18) in the super-hydrophobic state is dominated by the gas–solid interface. In addition, the decay distance of fluorescence intensity at the interface increases with the increasing of hydrophobicity, which is related to the penetrated H₂O concentration (\*H

coverage) on the surface. These results suggest that the variation of the interface contact state derived from the interfacial wettability affects both CO₂ and H₂O transport, which may be a solution for simultaneously tuning kinetic-controlled \*CO and \*H.

The local CO₂/H₂O ratio derived from wettability may affect the coverage of \*CO and \*H, resulting in the reaction pathways toward ethylene or ethanol. CFD simulation is adopted to understand the effect of wettability on the interfacial contact state and the local CO₂/H₂O ratio, which can reveal the mechanism of wettability variation influence on the ethylene and ethanol pathways (Fig. 4g–j). As shown in SEM images, Cu nano-islands are uniformly grown on the porous surface of carbon paper with an average size of around 1 μm (Supplementary Figs. 2 and 3). Therefore, the structure of the Cu electrode can be simplified as catalyst islands with pores. To enable the CFD simulation, a simplified model was constructed (Fig. 4g). The squares in the model are used to represent the catalysts with various contact angles. The gaps between the squares represent the pores between the real catalyst islands (Fig. 4g). The simulation results reveal the formation of a continuous liquid layer on the hydrophilic catalyst (Cu) surface and CO₂ needs diffuse through the thin liquid layer to reach the catalyst for reduction, which causes some hinder during gas transport (Fig. 4h). In comparison, the whole surface of high hydrophobicity (CA:156°) catalyst is mostly occupied by gas phase and formed a gas–solid interfaces. The hindrance of gas transport through the thin liquid layer is eliminated, which will block H₂O transport from the electrolyte, simultaneously (Fig. 4j). In comparison, a gas–liquid–solid interface is formed on the lower hydrophobic catalyst (CA: 131°) with balanced wettability (Fig. 4i, j). The exposure of the three-phase interface may balance the gas and liquid mass transport, resulting in an optimized \*CO/\*H ratio for ethylene and ethanol conversion. These results show

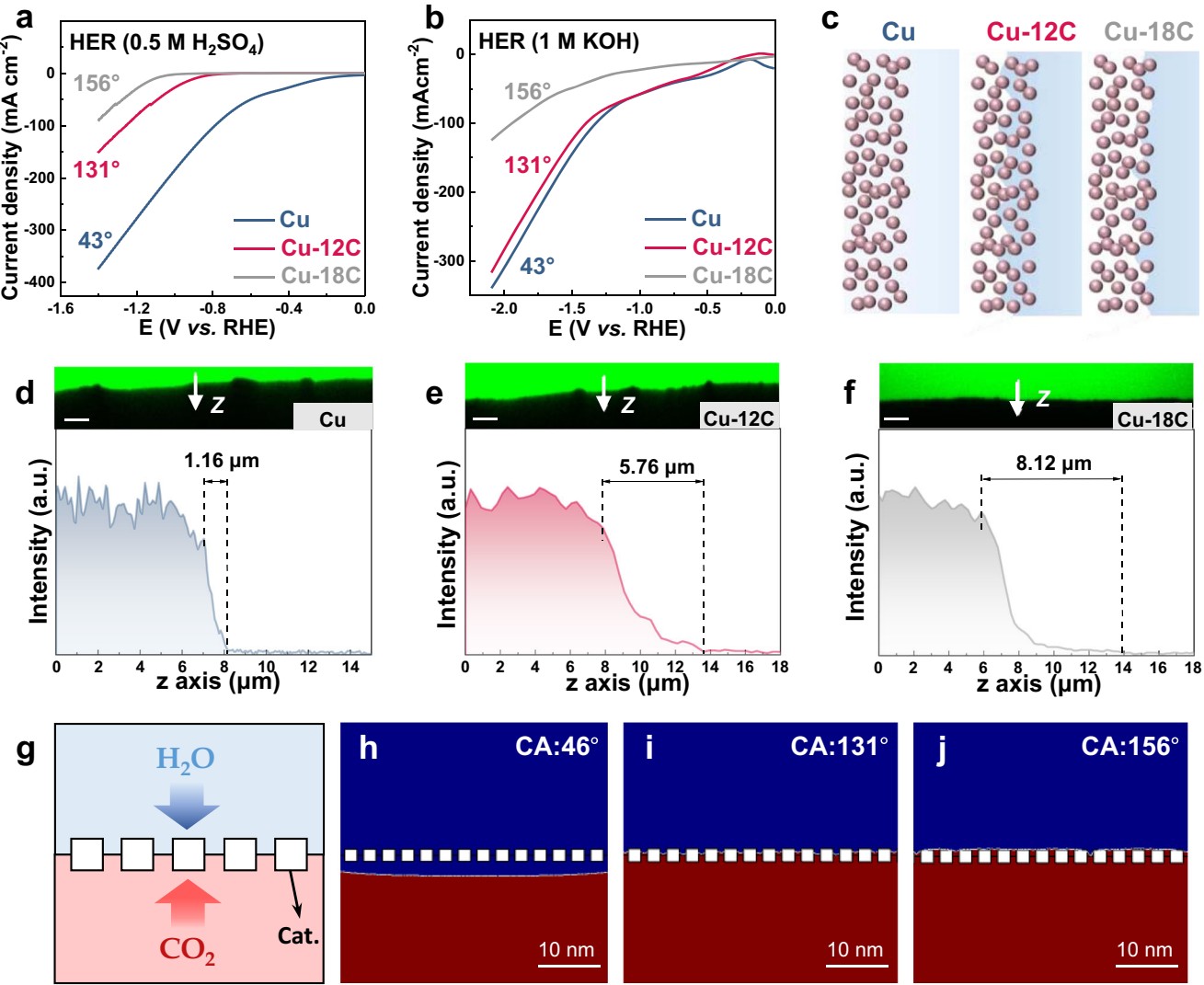

**Fig. 4 | Effect of controllable wettability on *H coverage via $H_2O$ mass transport.** **a** LSV curves of $H_2$ evolution on the Cu electrodes with different contact angles in 0.5 M $H_2SO_4$ electrolyte and **b** 1 M KOH electrolyte ($v$ = 50 mV s$^{-1}$). The hydrophilic Cu electrode (CA: 43°, blue line) exhibits optimal hydrogen evolution activity, suggesting efficient $H_2O$ transport in a hydrophilic environment. **c** Schematic illustration of available $H_2O$ concentration at reaction interface of Cu, Cu-12C, and Cu-18C, respectively. The white and blue parts in the schematic represent gas and liquid, respectively. The balls represent catalysts. The catalyst layers in the hydrophilic (Cu) and super-hydrophobic state (Cu-18C) are dominated by the liquid-solid interface and the gas-solid interface, respectively. The catalyst layer with balanced wettability (Cu-12C) is occupied by gas-liquid-solid interface. **d–f** Cross-sectional fluorescence images (scale bar: 20 μm) and corresponding $z$-axis fluorescence intensity line scans of labeled regions (white arrows) of Cu, Cu-12C, and Cu-18C,

respectively. The decay distance of the fluorescence intensity increases with the improvement of the hydrophobicity of the catalyst layer, indicating the smaller available $H_2O$ concentration. **g** Schematic of the CFD simulation. The red and blue parts in the schematic represent gas and liquid, respectively. The squares represent catalysts. The gaps between the squares represent the pores between the real catalyst islands. The different wettability of the catalyst layer will lead to differences in the interfacial contact state and the local $CO_2/H_2O$ ratio. Cat. catalyst. **h–j** Comparison of the modeled gas–liquid mass transport with different interfacial wettability, in which the red color in the simulation result represents a $CO_2$ volume fraction of 100% (electrolyte 0%), while the blue color represents a $CO_2$ volume fraction of 0% (electrolyte 100%). The $CO_2/H_2O$ ratio at the reaction interface increases with the enhance of hydrophobicity. CA: contact angle. The a.u. stands for arbitrary units.

that the spreading of gas and electrolyte on the catalyst surface can be controlled by wettability. A gas–liquid–solid interface may balance gas and liquid mass transfer to achieve a suitable *CO/*H ratio for producing ethylene and ethanol.

### Illustration of the role of controllable wettability on the reaction pathways of ethylene and ethanol

Based on the aforementioned, the mechanism of interfacial wettability effect on the ethylene and ethanol pathways can be deduced (Fig. 5). *CO coverage and *H coverage has been widely accepted as a crucial factor in the selective production of ethylene or ethanol[5–8]. Namely, the *CO and *H intermediates on the catalyst surface can affect the pathways of ethylene and ethanol. Inspired by previous works[5–8], we

speculate that the variation of *H and *CO coverage caused by the local concentration of $CO_2$ and $H_2O$ via wettability may be one of the important reasons for the selectivity changes in ethylene and ethanol. It is worth noting that altering *CO/*H surface coverage not only affects ethylene and ethanol formation but also affects other products. Herein, only the main products, namely ethylene and ethanol, are discussed. In general, with the increasing of hydrophobicity, the interfacial structure shifts from liquid–solid interface (CA: 43°) to gas–liquid–solid interface (CA: 131°) and then to gas–solid interface (CA: 156°) (Fig. 5a). Different interfacial structures influence the mass transport of $CO_2$ and $H_2O$, which may lead to the variation of *CO and *H coverage. Moreover, the voltage ranges from in-situ ATR-SEIRAS spectra and the decay distances from CLSM was adopted to quantity

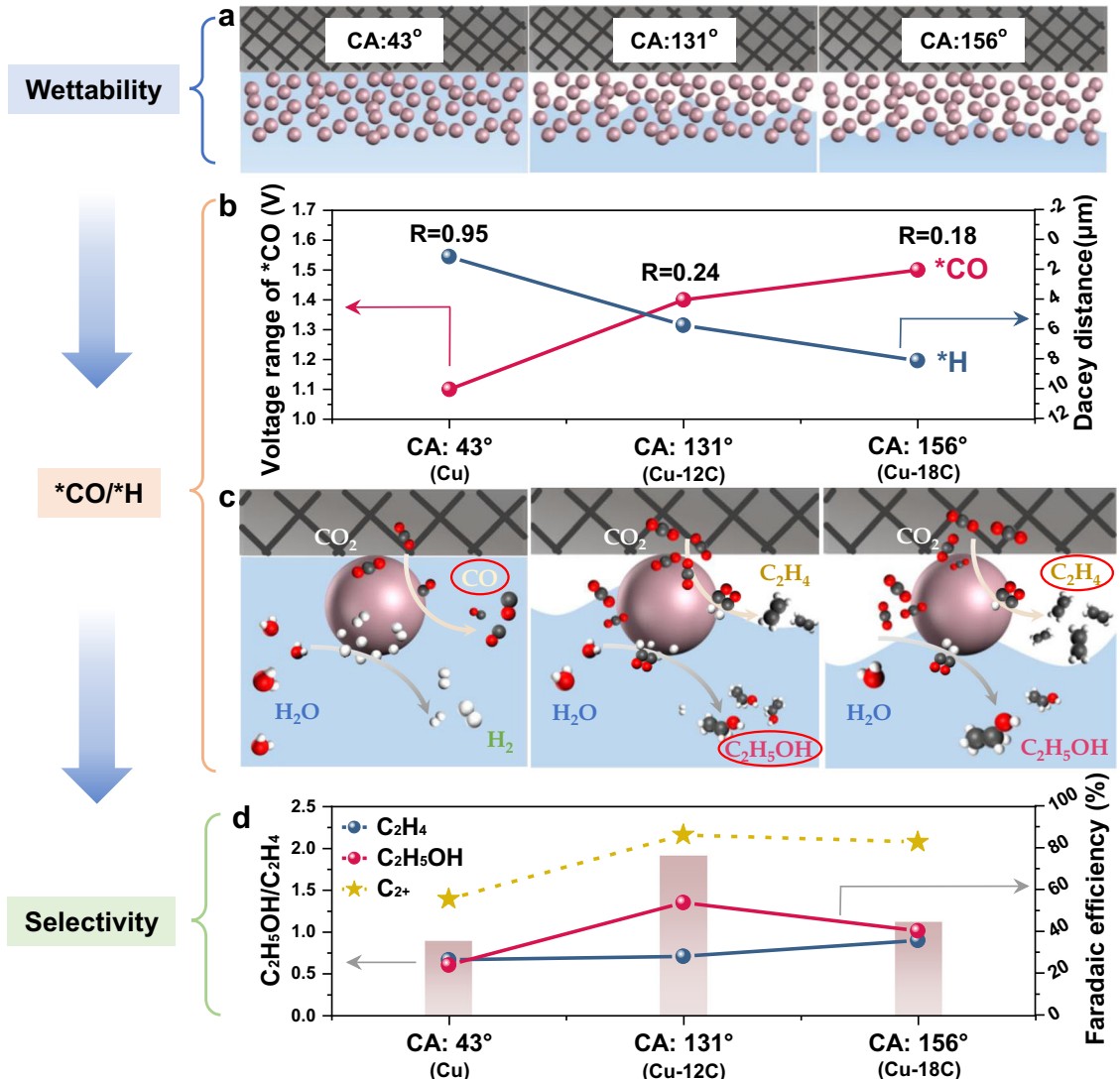

**Fig. 5 | Illustration of the role of controllable wettability on the reaction pathways of ethylene and ethanol. a** Schematics of interfacial contact states of Cu (CA: 43°, liquid-gas contact), Cu-12C (CA: 131°, gas–liquid–solid contact), and Cu-18C (CA: 156°, gas-solid contact). The white and blue parts in the schematics represent gas and liquid, respectively. **b** Correlation between interfacial wettability and *CO and *H. Voltage ranges from in-situ ATR-SEIRAS spectra and decay distances from CLSM are used to compare *CO and *H coverage, respectively, at different interfacial wettability. With the increase of hydrophobicity, *CO coverage increases while *H coverage decreases. The ratio of voltage range (*CO) over decay distance (*H) (denoted as R, V/μm) is adopted to represent the *CO/*H ratio. **c** Schematics of the reaction pathways and product formation on Cu, Cu-12C, Cu-18C with different wettability (blue: liquid, white: gas). **d** The variety of product selectivity with different interfacial wettability. The reaction pathway of $C_2$ product is jointly determined by *CO coverage and *H coverage. The Faradaic efficiencies toward ethanol and $C_{2+}$ of up to 53.7% and 86.1%, respectively. The illustrations show that the kinetic-controlled *CO/*H ratio derived from the interfacial contact state affects the ethylene and ethanol pathways. CA: contact angle.

the changes of *CO and *H coverage. In general, a wider *CO adsorption voltage indicates a higher *CO coverage in in-situ ATR-SEIRAS spectra, while a larger decay distance of fluorescence intensity via CLSM represents a lower *H coverage. The experimental data reveal that the increase of hydrophobicity will lead to an increasing *CO coverage and a decreasing *H coverage (Fig. 5b).

Further, the reaction pathways on catalyst surfaces with different wettability can be understood (Fig. 5c). At equilibrium, the surface coverage of the intermediate (*CO, *H) is directly proportional to the concentration of reactants ($CO_2$, $H_2O$). For example, the relevance between the surface coverage of adsorbed $*CO_x$ ($\theta_{CO_x}$, $x=1$ or 2) on the catalyst and the local concentration of $CO_x$ ([$CO_x$]), which is given as Eq. (1)[5,42]

$$\theta_{CO_x} = \theta \times [CO_x] \times e^{-\frac{E_{CO_x}}{RT}} \qquad (1)$$

where $\theta$ is the coverage of free surface sites, $E_{CO_x}$ is the $CO_x$ adsorption energy on the surface, $R$ is the ideal gas constant and $T$ is the temperature. The coverage of intermediate ($\theta_{CO_x}$) is a function of CO adsorption energy ($E_{CO_x}$) and local $CO_x$ concentration ([$CO_x$]). To vary the intermediate (*CO/*H) coverage, changing the local concentration of reactant ($CO_x$) at the catalyst layer rather than *CO/*H adsorption energy is, therefore, a promising approach. Compared with the classical modulation of the active site, the novelty of our method is that the $CO_2RR$ is completely controlled by kinetics (mass transport) through controllable interfacial wettability (Supplementary Figs. 39, 40). Thus, the hydrophobic modification can enhance the gas transport and increase the surficial adsorption of $CO_2$, resulting in the increase of $*CO_2$ intermediate under bias. Moreover, the hydrophobic surface may further trap the produced CO, which can increase the local concentration of CO and the coverage of *CO intermediate, thus enhancing the C–C coupling to $C_{2+}$ products[5,20,48]. Another key intermediate

*H is converted from the bulk solution. As analyzed above, the *H coverage can be deduced indirectly through water content. Thus, a hydrophobic surface can hinder water absorption and lower *H coverage. As suggested by recent reports[7,8,50], the higher *H coverage and *H transport efficiency can benefit the generation of alcohol. Therefore, the construction of gas–liquid–solid interface through creating a surface with suitable hydrophobicity can increase the selectivity of ethanol, while a super-hydrophobic surface will lead to ethylene production. Therefore, it is speculated that the reaction pathways of ethylene and ethanol can be tuned by kinetic-controlled *CO/*H ratio through controllable wettability, which is enabled by modifying alkanethiols with different alkyl chain lengths.

Similar phenomena can be observed by the varying selectivity with different interfacial wettability (Fig. 5d). The Faradaic efficiencies of $C_{2+}$ and ethanol are increased first and then decreased with the increasing of hydrophobicity, which reveals the $C_{2+}$ reaction pathways can be modulated by balancing the mass transport of $CO_2$ and $H_2O$. To tentatively quantify the balance between $CO_2$ and $H_2O$ mass transport, the ratio of voltage range (*CO) over decay distance (*H) (denoted as R, V/μm) is adopted to represent *CO/*H ratio. Generally, a higher R-value indicates a lower ratio of *CO/*H. With the increasing of hydrophobicity, R-value gradually decreases from 0.95 to 0.24 and then to 0.18, indicating the increased *CO/*H ratio. The selectivity of ethanol and $C_{2+}$ products can reach the maximum through balancing $CO_2$ and $H_2O$ mass transport with a R-value of 0.24. Therefore, with the increasing of the hydrophobicity, the coverage ratio of *CO/*H may increase simultaneously. The limitation step of $CO_2$ reduction is also change from insufficient *CO to inadequate *H, which may lead to changes in the distribution of ethylene and ethanol.

## Discussion

A direct and efficient interfacial modification protocol has been developed to continuously modulate the wettability of the catalyst. Through changing the alkyl chain lengths on alkanethiols, the equilibrium of kinetic-controlled *CO and *H intermediates on the surface may be controlled. In the flow cell, continuous modulating $CO_2$ reduction to produce ethanol and ethylene with the ratio arranged from 0.9 to 1.92 is achieved, with the highest Faradaic efficiencies of 53.7% and 86.1% for ethanol and $C_{2+}$, respectively. A $C_{2+}$ Faradaic efficiency is up to 80.3% can be maintained even at a higher current density of 400 mA cm$^{-2}$, corresponding to a $C_{2+}$ partial current density of 321 mA cm$^{-2}$, which is a remarkable advance over the recently reported Cu electrode. Moreover, the models have been built to exploit the relevance of reaction pathways with *CO/*H coverage ratio, which is derived from ATR-SEIRAS spectra and decay distances from CLSM. Interfacial hydrophobic treatment may accelerate $CO_2$ mass transport while hindering $H_2O$ absorption, resulting higher *CO/*H ratio. The reaction limitation may shift from *CO insufficiency to *H exhausting, resulting in the main product changes from ethanol to ethylene. Controllable wettability modulation protocol here can elaborately balance the $CO_2$ and $H_2O$ mass transport, which specially synthesizes $C_{2+}$ product with highly selective in electrode design. Moreover, future research endeavors may focus on promoting moderate and stable hydrophobicity on porous electrodes to further boost activity and selectivity.

## Methods

### Preparation of Cu electrode

Cu catalyst (Cu target, 99.999%, Zhongnuo Advanced Material Technology Co., Ltd.) was deposited onto carbon paper gas diffusion layers (2 × 2 cm$^2$, Sigracet 29BC, SGL Carbon) through a direct current (DC) magnetron sputtering system. The base pressure was $2.0 \times 10^{-4}$ Pa, and the flow rate of Ar was set as 20 standard cubic centimeters per minute (sccm). The DC power was 40 W, and the working pressure was 1 Pa. The deposition time was 10 min.

### Preparation of Cu-4C, Cu-7C, Cu-12C, Cu-18C

The Cu electrode was submerged into the 1-dodecanethiol (liquid, 98%, Aladdin Co. Ltd.) under Ar and left for 1 min. The formation of the thiol layer (Cu-xC) was performed in pure thiol liquid to ensure that thiol molecules reach saturated adsorption on the copper surface. After this, the electrode was moved to a solution of ethyl acetate (99.5%, Aladdin Co. Ltd.) for 5 min to remove residual 1-dodecanethiol and then was dried under vacuum at 60 °C, and the resultant electrode was denoted as Cu-12C. The Cu-4C, Cu-7C can be obtained by the same method from 1-butanethiol (97%, Aladdin Co. Ltd.) and 1-heptanethiol (98%, Aladdin Co. Ltd.), respectively. As for Cu-18C, 1-octadecanethiol (97%, Aladdin Co. Ltd.) was first melted under a vacuum at 60 °C, and the Cu electrode was submerged into the 1-octadecanethiol under Ar and left for 10 min. Then, the electrode was moved to a solution of ethyl acetate at 60 °C for 5 min to remove residual 1-octadecanethiol and was dried under vacuum at 60 °C.

### Preparation of CuAg electrode

Ag catalyst (Ag target, 99.999%, Zhongnuo Advanced Material Technology Co., Ltd.) was deposited onto the Cu electrode through DC magnetron sputtering system. The base pressure was $2.0 \times 10^{-4}$ Pa, and the flow rate of Ar was set as 20 sccm. The DC power was 10 W, and the working pressure was 1 Pa. The deposition time was 10 s.

### Characterization

The crystal structure was determined by X-ray diffractometer (XRD, Bruker D8 Focus) with Cu Kα radiation ($\lambda = 1.54056$ Å), XPS analyses were carried out on a Physical Electronics PHI 1600 ECSA system with an Al Kα X-ray source (1486.6 eV). Field emission scanning electron microscopy (FESEM) images were carried out on Hitachi S-4800 with an acceleration voltage of 5 kV. Transmission electron microscopy (TEM) images were performed on JEOL JEM-2100F, using the Tecnai G2 F20 microscope with an acceleration voltage of 200 kV. Fluorescence spectra were obtained on F-4600 (Hitachi, Japan) luminescence spectrometer. Confocal laser scanning microscopy (CLSM) images were performed on Nikon A1 (Japan). The contact angles were measured on the Powereach JC2000C1 contact angle system.

### Electrochemical measurements

$CO_2$ reduction was conducted in a three-chamber flow cell manufactured by Gaossunion Co., Ltd. (Supplementary Fig. 8), where the $CO_2$ gas was supplied directly to the catalyst layer (cathode, working electrode). Ni foam (0.5 cm × 1.0 cm, Tianjin Incole Union Technology Co., Ltd.) and saturated Hg/HgO electrode (Gaossunion Co., Ltd.) were used as the counter electrode (CE) and the reference electrode (RE), respectively. The cathode and anode compartment was separated by an anion exchange membrane (AEM, FAA-3-PK-75, Fumatech), both of which were filled with 1 M KOH electrolyte. $CO_2$ (99.995%, Air Liquide Co. Ltd.) flowed into the cell at a flow rate of 20 sccm (standard cubic centimeter per minute) controlling by a mass flow controller (MC-Series, Alicat Scientific). To ensure the accuracy of the product selectivity calculation, another flow meter (MC-Series, Alicat Scientific) was used to accurately measure the outlet $CO_2$ flow rate. Peristaltic pumps (EC200-01, Gaossunion Co., Ltd.) were used to control the flow rate of the electrolytes at -10 mL min$^{-1}$. The gas products were quantified by gas chromatography (GC) (Ruimin GC 2060, Shanghai), which is online connected with the flow cell. CO and $H_2$ were detected by a flame ionization detector (FID) and thermal conductivity detector (TCD), respectively. The liquid products were quantified by using static headspace gas chromatography (HS-GC, Agilent Technologies, 7890B). All potentials were not iR corrected. During the CORR experiment, a continuously flowing $CO/N_2$ mixed gas (CO: 10 sccm; $N_2$: 20 sccm) was directed into the gas compartment.

## Wettability measurements

A sessile drop method was used for wettability tests by a contact angle system (Powereach JC2000C1, Shanghai Zhongchen Digital Technology Apparatus Co., Ltd.) at ambient temperature. The water droplet with a volume of 10 mL spread on the sample surface over time. Contact angles were estimated after stabilizing the water droplet on the surface. The average contact angle was obtained by averaging the data in three independent measurements.

## $OH^-$ electroadsorption measurements

In-situ $OH^-$ electroadsorption measurements were performed in a flow cell and a potentiostat (Autolab PGSTAT 204, Metrohm). Ni foam (Tianjin Incole Union Technology Co., Ltd.) was used as the counter electrode, and saturated Hg/HgO electrode (Gaossunion Co., Ltd.) as a reference electrode. An aqueous solution of 1 M KOH was used as the electrolyte. Before electrolysis, cyclic voltammetry (20 mV/s) was performed by flowing Ar. Then, $CO_2$ reduction was conducted at −1.2 V vs. RHE for 6.5 h in a flow cell with $CO_2$ feeding. After electrolysis, the $CO_2$ feeding was immediately switched to Ar feeding, and then cyclic voltammetry (20 mV/s) was performed after the electrolyte stopped flowing[12,51].

## Electrochemical impedance spectroscopy (EIS)

EIS measurements were performed in a flow cell at room temperature with Autolab electrochemical workstation. Ni foam (Tianjin Incole Union Technology Co., Ltd.) and saturated Hg/HgO electrode (Gaossunion Co., Ltd.) were used as the counter electrode and the reference electrode, respectively. The EIS measurements were carried out in 1 M KOH aqueous solution at open circuit potential (OCP). The impedance spectra were recorded with an amplitude from 10 mV of 0.01–100 kHz. The data obtained from the EIS measurements were fitted by the Zview software (Version 3.1, Scribner Associates, USA).

## ECSA measurements

The ECSA was measured by the double-layer capacitance ($C_{dl}$) of electrodes in 1 M KOH electrolyte in a flow cell. Firstly, $CO_2$ electroreduction was conducted at −1.2 V vs. RHE for 1 h. After electrolysis, the $CO_2$ feeding was immediately switched to Ar feeding, and then cyclic voltammetry was performed after electrolyte stopped flowing. The scan rate ranged from 20 to 120 mV s$^{-1}$. The function of current and scan rate was established to determine the $C_{dl}$. ECSA was obtained by normalizing the $C_{dl}$ with that of a Cu foil[12].

## Brunauer–Emmett–Teller (BET)

A Micromeritics Tristar 3000 analyzer was used to obtain the textual properties of catalysts by $N_2$ adsorption-desorption at 77 K. Prior to the tests, all samples were degassed at 300 °C for 6 h. It is difficult to obtain enough BET samples by DC sputtering. Thus, the copper powder with and without alkanethiol treatment was used to verify the influence of alkanethiol modification on Brunauer–Emmett–Teller-specific surface area.

## In-situ ATR-SEIRAS measurements

Au film was deposited directly on a Si prism according to the previous work reported by Dunwell et al.[52]. The Si prism was firstly polished by the $Al_2O_3$ powder (0.05 µm, Gaoss Union Technology Co. Ltd.) to a hydrophobic surface. Then, isopropanol (99.7%, Tianjin Yuanli Chemical Co. Ltd.) and deionized water were used to wash the Si prism to remove the residue and $Al_2O_3$ powder. Subsequently, the Si prism was immersed in the mixture of $H_2SO_4$ (95−98%, Sigma-Aldrich) and $H_2O_2$ (30%, Sigma-Aldrich) with a volume ratio of 3:1 for 20 min to remove organic contaminants. To improve the adhesion of the Au film, the reflecting plane of the prism crystal was immersed in 40% $NH_4F$ solution (Sigma-Aldrich) for 90 s to remove the oxide layer and generate a hydrogen-terminated surface. The reflecting plane of the Si prism was then immersed in a volume mixture (4.4:1) of 2% HF and Au plating solution consisting of 5.75 mM $NaAuCl_4·2H_2O$, 0.025 M $NH_4Cl$, 0.075 M $Na_2SO_3$, 0.025 M $Na_2S_2O_3·5H_2O$, and 0.026 M NaOH for 5 min at 55 °C. After deposition, the Si prism was cleaned with deionized water carefully. Subsequently, the Cu catalyst was sputtered onto the Au surface, the alkanethiols were then dropped on the Cu surface.

In-situ ATR-SEIRAS was performed on the FT-IR spectrometer (is50, Nicolet) with a modified accessory at a 60° incident angle (VeeMax III, PIKE Technology). The operando spectroelectrochemical cell was designed according to the previous work reported by Li et al.[53]. Pt foil and Ag/AgCl electrode (Gaossunion Co., Ltd.) was used as the counter electrode and reference electrode, respectively. An anion exchange membrane (AEM, FAA-3-PK-75, Fumatech) was used to separate the cathode and anode. Before performing the experiment, 5 sccm (standard cubic centimeter per minute) $N_2$ was used to purge the optical path system for 5 h to reduce the impact of $CO_2$ (g) and $H_2O$ (g) in the air. The background was taken in an Ar-saturated electrolyte for each electrode. Later, 20 mL of electrolyte was bubbled with $CO_2$ for 30 min before the test. Electrochemical measurements were conducted on a potentiostat (CompactStat.e20250, IVIUM), and all the spectra were collected at the resolution of 4 cm$^{-1}$ and 8 scans[12].

## In-situ Raman spectroscopy measurements

In-situ Raman spectroscopy was carried out in a custom-designed flow cell with a potentiostat (CompactStat.e20250, IVIUM). The geometric surface area of the working electrode was 1 cm$^2$. A Pt wire and an Ag/AgCl electrode (Gaossunion Co., Ltd.) were used as the counter and the reference electrode, respectively. The cathode and anode were separated by an anion exchange membrane (AEM, FAA-3-PK-75, Fumatech). An aqueous solution of 1 M KOH was used as the electrolyte. In situ, Raman spectroscopy was performed on a Raman microscopy system (LabRAM HR Evolution, Horiba Jobin Yvon). A 785 nm laser was used as the excitation source[12].

## Fluorescence electrochemical spectroscopy (FES)

A fluorescence spectrophotometer (F-4600, Hitachi, Japan) coupled with an electrochemical workstation (CompactStat.e20250, IVIUM) in the in-situ FES system. 8-hydroxypyrene-1,3,6-trisulfonic acid (HPTS) was used as a pH-sensitive fluorescence probe molecule here for the determination of the interfacial $CO_2$ concentration (Supplementary Fig. 28). Excitation and emission wavelengths for HPTS were 485 and 520 nm, respectively. The intensity of the HPTS characteristic emission peak shows a strong $CO_2$ concentration dependence. A time scan PL intensity curve and a fluorescence intensity versus $CO_2$ concentration standard curve were collected without electrolysis. Specifically, the total gas flow rate was controlled at 50 sccm (standard cubic centimeter per minute) by using a mass flow controller (MC-Series, Alicat Scientific) during the measurement. The $CO_2$ concentration in the gas phase was switched from 0% to 10% in 1% increments by using Ar as the carrier gas. Then, the fluorescence intensity could be correlated with $CO_2$ concentration (standard curve, Supplementary Fig. 29). Subsequently, according to the standard curve, the corresponding $CO_2$ concentration could be obtained by measuring the fluorescence intensity of different systems under electrolysis conditions[48].

## Confocal laser scanning microscopy (CLSM)

The variation of available water on the interface can be estimated by confocal laser scanning microscopy (Nikon A1, Japan) with the confocal microscope. 100 µL of fluorescein-labeled 1 M KOH was added to confocal dishes, and a 5 × 5 mm$^2$ Cu electrode was placed on the liquid droplet with the catalyst side in contact with the electrolyte[48]. The interfacial contact state of electrodes with different wettability can be directly reconstructed by the analysis of a series of CLSM images at different depths within the catalyst layer (Supplementary Fig. 37).

## Calculation of $CO_2$ conversion

The calculation of $CO_2$ conversion was given as following Eqs. (2) and (3)[54]:

$$\left(\frac{mC}{s\,cm^2}\right)\left(\frac{1C}{1000\,mC}\right)\left(\frac{mol\,e-}{96485\,C}\right)\left(\frac{1\,mol\,ethene}{12\,mol\,e-}\right)\left(\frac{2\,mol\,CO_2}{1\,mol\,ethene}\right)\left(\frac{22.4\,L}{1\,mol\,CO_2}\right)$$

$$\left(\frac{1000\,mL}{1\,L}\right)\left(\frac{60\,s}{1\,min}\right)(1\,cm^2\,geometric) \qquad (2)$$

$$= mL\,min^{-1}\,CO_2\,consumed\,by\,device\,for\,ethene\,synthesis$$

$$100 \times \left(\frac{mL\,min^{-1}\,CO_2\,consumed}{mL\,min^{-1}\,CO_2\,flowed\,in}\right) = \%\,CO_2\,conversion\,to\,ethene \qquad (3)$$

## Computational fluid dynamic (CFD) methods

The wettability of the electrode was varied in the simulations by varying the contact angle. The momentum equations for the entire flow field were given as Eqs. (4)–(7):

$$\rho \times \frac{\partial u}{\partial t} + \rho \times (u \times \nabla) \times u = \nabla \times [-P + \tau] + F + \rho \times g \qquad (4)$$

$$\rho \times \nabla \times u = 0 \qquad (5)$$

$$\tau = \mu \times \left(\nabla \times u + (\nabla \times u)^T\right) \qquad (6)$$

$$F = \sigma \times k \times n \times \delta \qquad (7)$$

where $\rho$ (kg/m$^3$) is the density, $u$ (m/s) is the velocity components in $y$ direction, $t$ (s) is the time, $P$ (Pa) is the pressure, $\tau$ is the deviatoric stress tensor for a Newtonian fluid, $F$ (N/m$^3$) is the surface tension force per unit volume, $g$ (m/s$^2$) is the acceleration due to gravity, $\mu$ (N s/m$^2$) is the dynamic viscosity, $T$ (K) is the temperature, $\sigma$ (N/m) is the surface tension coefficient, $k$ (m$^{-1}$) is the interfacial curvature, $n$ is the interfacial unit normal vector, and $\delta$ is the delta function centered at the interface.

## Density functional theory (DFT) methods

The theoretical calculation was conducted by Vienna ab initio simulation package (VASP) with the BEEF-vdW exchange-correlation functional[55]. As for the simulation models, we built five-layer Cu(111)-(4 × 4) slabs with a Butyl Mercaptan and 1-dodecanethiol molecular to compare the effect of different lengths of thiols on the reaction mechanism. Three bottom layers were fixed while the upper layers were relaxed in these models. The periodic interactions between repeated slabs were minimized by the vacuum space of at least 15 Å. As for the calculation settings, the cut-off energy is 400 eV. The interactions between the atomic cores and electrons were described by the projector augmented wave (PAW) method[56]. All structures were optimized until the force on each atom has been <0.02 eV/Å. A (3 × 3 × 1) $k$-point grid was employed for the simulation models.

The free energy of the *CO and *H ($\Delta G_{ads}$) was calculated as following Eqs. (8) and (9):

$$\triangle G_{*CO} = G_{*CO} - G_{CO(gas)} - G_{surface} \qquad (8)$$

$$\triangle G_{*H} = G_{*H} - 1/2G_{H_2(gas)} - G_{surface} \qquad (9)$$

## Data availability

The data generated in this study are provided in Supplementary Information and Source Data file. Source data are provided with this paper.

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

## Acknowledgements

We acknowledge the National Key R&D Program of China (2021YFA1500804), the National Natural Science Foundation of China (22121004, 51861125104, 22038009, and 22250008), the Natural Science Foundation of Tianjin City (18JCJQJC47500), Haihe Laboratory of Sustainable Chemical Transformations (CYZC202107), the Program of Introducing Talents of Discipline to Universities (No. BP0618007), and the Xplorer Prize for financial support.

## Author contributions

J.G. supervised the project. J.G., T.W., and Y.L. conceptualized the project. Y.L., G.Z., and Q.C. prepared electrodes. Y.L., G.Z., Q.C., and H.G. conducted the catalytic tests and the related data processing. C.P., Z.-J.Z., L.Z., L.L., Y.L. K.H., and Z.P. carried out the theoretical calculations. All authors participated in the discussion of the research.

## Competing interests

The authors declare no competing interests.
