## [Peer Review File · Nature Communications]

REVIEWER COMMENTS

Reviewer #1 (Remarks to the Author):

In this work, Lin and coworkers present a mechanistic study on the influence of cathode surface wettability on the CO₂ reduction reaction pathways towards ethanol vs. ethylene. The authors modify the surface of Cu gas diffusion electrodes with alkanethiol self-assembled monolayers with different lengths. With increasing SAM length, the electrodes exhibit an increased hydrophilic behavior. The authors correlate the different surface properties with the different selectivity towards ethanol and ethylene, presenting a mechanistic picture on the role of H₂O, CO_x and H transport and interaction.

The work is interesting and could be suitable for this journal provided some important points were considered.

-SH SAM formation

Could the authors comment on the presence (and role) of Cu²⁺ on bare Cu samples (lacking in the -SH functionalized electrodes). Do these results imply that -SH bonding occurs through previously oxidized Cu sites? On the XPS results, do the authors observe any shift or artifact due to the lower conductivity of -SH samples?

What would then be the impact once the samples reduce at cathodic potential? This could lead to different reconstruction processes, and different Cu facets, which might affect local H and CO adsorption and selectivity.

More details are needed on the SAM incubation. It seems that the formation of the thiol layer proceeds in saturated thiol solutions? (or is a mM concentration fixed). This would lead to uncontrollable coatings, limited by hindrance and surface competition. E.g., a different mols/nm² loading of coated molecules for different number of carbons. While this might not affect the ultimate hydrophilic, it could have implications on the surface accessibility of reactants, and other electronic properties.

At which voltages/pH conditions do the authors expect the SAM to be stable or reduced?

Mechanism analyses

The reported JV curves exhibit a clearly different behavior for samples coated with the SAMs. The authors perform most comparative analysis at a fixed potential vs RHE. This would be fair in terms of thermodynamics (the catalysts and reactants would face ~similar barriers), but would neglect important kinetic and dynamic environment properties such as OH coverage, pH etc. The authors are encouraged to perform similar analysis at fixed currents to offer a more complete vision on the proposed mechanisms.

On a same note, the authors report different ECSA for SAMs with different lengths. This could be the case considering the previous point (e.g., different loading and less active sites that are occupied by SH bonds). Another important aspect is that the samples might exhibit a different in plane and out of plane resistance. This would complicate ECSA analysis based on double layer capacitance (which assumes a similar R). A proper ECSA analysis is also important considering the dynamic surface competition. Detailed EIS analyses or other techniques would help clarify these points.

Modelling

It is not clear to this reviewer that Fig 5 is representative to the actual catalysts or is adding any value. e.g., the surface of the Cu electrode would be continuous and gas flow and partial CO₂ pressures would be different.

The proposed picture in Fig 6 is nice but might be misleading. Have the authors proven that for the EtOH and C₂H₄ reactions CO₂ is coming through the gas phase (and not dissolved in the electrolyte as mentioned in the intro)

Performance

Stability should be reported and sufficient to sustain the validity of all the studies presented herein. It is important to see if the SAM functionalization would affect stability

What is the single pass carbon utilization?

Others

What is the thickness of the Cu electrodes? This is not clear in the methods section

Elaborate more on the novelty and new findings compared to Nature Materials volume 18, pages 1222–1227 (2019) and other works employing a similar approach.

Language and typos (e.g., ESCA vs ECSA) should be fixed

Reviewer #2 (Remarks to the Author):

Lin and Wang took an interesting approach to control the surface hydrophobicity of copper catalyst and investigate its impact on the reaction mechanism in the CO₂RR, especially towards ethanol and ethylene formation. They aim to understand how tuning the surface wettability affects the reaction pathway via adjusting the *CO and *H surface coverage. This is an important topic to be discussed in literature. The experiments are mostly designed well, however, the results are not fully supporting the hypothesis in this work. This reviewer believes that the submitted work need to be further improved before being confident to recommend it for publication. The main questions to be answered by the authors are listed below:

- Effect of wettability on *CO coverage is investigated experimentally using ATR-SEIRAS, but there is no direct experimental evidence to show its impact on *H coverage. The HER experiment in Fig. 4 cannot conclusively prove that we have a different *H coverage on the surface. Perhaps Raman would be useful here.
- The role of wettability on *CO and *H coverage is shown (if we accept the discussion for the *H coverage for now), but still, it is not clear why and how these surface coverages affect the reaction pathway towards ethylene and ethanol? That is a key question which is stated in the introduction but not fully answered in this paper. To be more specific, why we have ethanol, and no other alcohols or oxygenates like methanol, propanol or acetate? Why this surface wettability affects ethylene and no other hydrocarbons like methane?
- In this work it is shown that, the higher *CO coverage is expected when the surface is more hydrophobic. And it's shown that the more hydrophobic surface such as Cu-12C is more selective to produce ethanol compared to ethylene. But then, why in CORR experiment we see more ethylene than ethanol compared to the CO₂RR experiment? Should not it be the other way around? I would expect a higher *CO surface coverage in CORR compared to that in CO₂RR, and as it's shown via ATR_SERIRAS test the higher *CO is the consequence of the more hydrophobic surface and logically we should expect more ethanol.

- The potential impact on the reaction pathway is important. By saying potential, we mean the potential on the surface of the catalyst. However, ignoring the iR potential loss cannot give us a realistic picture of what potential exists at the cathode. Therefore, the whole discussion in this manuscript can go under question that perhaps all observations are due to the potential shift, and that's why we see different product distribution. EIS test must be performed to measure the impedance of the system and also to elaborate more on the charge transfer resistance for each catalyst.
- In the introduction, one of the main challenges in the field is discussed as ethylene/ethanol selectivity control, however, throughout the manuscript, ethylene is replaced by C₂⁺ products invariantly which makes the story difficult to follow and to draw a clear conclusion.
- To further support the hypothesis that the surface wettability is the only parameter affecting the selectivity (at least in this study), the authors need to repeat the experiment with a different control sample with a given hydrophobicity. For example, applying PTFE on the surface, measuring the contact angle, running the CO₂RR, and monitoring the ethanol/ethylene ratio.

A few minor comments:

- Line 120: nano-features are not visible in Supplementary Fig. 2. The scale bar in SEM images is in micro scale.
- Fig. 3c: the results for Cu-12C are not shown.
- There are several typos, grammatical mistakes, and difficult sentences to read, even wrong figure numbers (most Fig. 2's must be Fig. 3's). The manuscript needs to be proofread again before resubmission.
- Line 129-130: a table of deconvoluted peaks for XPS results could be provided in SI. Also, a survey can be added.
- Line 129-130: For Cu XPS results, did authors etch the surface of the sample? Cu surface can be easily oxidized (to Cu²⁺) in the atmosphere, and that's why they observed a little amount of Cu²⁺ for the Cu sample. This reviewer would suggest etching the surface of the sample first, then running XPS for a better comparison.
- Line 135-136: It is believed that adding different alkyl chain changes the contact angle. Is the alkyl chain the only parameter to change the contact angle? Or the amount of alkyl chain is also important. For example, if we have two samples: (1) Adding 1 ml of 12C, (2) Adding 5 ml of 12C, do we get the same contact angle? If it has not been done, a control test like that is suggested.
- Fig. 2: A table with exact FEs for all products at different contact angles should be provided in the SI.
- Fig. 2e is confusing. It is better to have the same range for the y-axis.
- Line 177: for the stability test, the authors only showed current density. Product distribution is also important and should be mentioned.

- Supplementary Fig. 7: there are some missing products in these figures, including methane, formate, and acetate. Those products should be added to make sure the total FE is around 100%.
- Supplementary Fig. 10: the caption is incorrect. Cu-xC, not Cu-12C.
- Supplementary Fig. 11: the FEs should be added to show the stability of the system.
- Supplementary Fig. 19: is the geometric surface area, correct? Almost 8m²/g?
- While using physical vapor deposition method, it is important to mention the thickness as it affects surface facets and other morphologic characteristics. This information is missing.
- Cite a few papers to support line 219.
- Add the synthesis details for the CuAg catalyst.
- In addition to the CFD simulations, density functional theory (DFT) computations can leverage the quality of the work, especially by looking into the adsorption energy (or if possible, the reaction pathways) of *CO and *H on different Cu-xC catalysts.

Reviewer #3 (Remarks to the Author):

The study investigates CO₂RR using different modifiers. There are several works currently published on this topic, however I do find parts of this work interesting. But I have multiple comments:

Please clarify what the authors mean when they put asterisk * on their molecules “*CO and *H”. The typical convention is that * means that CO is adsorbed on the surface.

The discussion of, or the framing of, *CO/*H feels a bit out of place. As a GDE does not change the binding energies of *CO or *H, but it rather changes the diffusion or chemical potential at the interface of the species, resulting in an improved performance of a GDE over a normal aqueous test. I believe the authors should be more clear on this point – as now this is mixed together. One example is:

“This work describes the design and realization of a strategy that continuously tunes the controllable equilibrium of *CO and *H by altering the interfacial wettability via alkyl thiols with different alkyl chain lengths.”

There are several sentences/statements with misleading references which needs to be corrected. Particularly in the introduction, I give some examples here:

“*CO and *H are considered to be the key intermediates during producing C₂+ products” Indeed *CO and *H are key intermediates for a predictive scheme for CO₂RR as first shown by Bagger et al. <https://doi.org/10.1002/cphc.201700736>

“As the only metal with a negative adsorption energy for *CO but a positive adsorption energy for *H, copper (Cu) presents a unique ability to produce C₂+ products^{9,10}. However, multiple products have been detected on Cu surfaces resulting in poor product selectivity for Cu¹¹.”

Please refer to the original and first publications by Bagger et al. <https://doi.org/10.1002/cphc.201700736> and <https://doi.org/10.1021/acscatal.9b01899>

“Previous investigations have found that the coverage of *CO and *H on Cu surface plays a critical role on determining selectivity of C₂+ products¹².” Ref 12: Energy Environ. Sci. 5, 7050–7059 (2012). ref 12 does not discuss *CO and *H on the Cu plays a critical role.

CO₂RR experiments carried out in 1M KOH. Fig 2 “e, Faradaic efficiencies of ethanol and ethylene on Cu electrodes in 1 M KOH (aq.) at –1.2 V versus the RHE with various contact angles (without iR correction).” When KOH is saturated with CO₂, it is not KOH anymore, but rather (bi)carbonate solution. Please clarify.

Fig2 f, clearly the Cu-12C and Cu-18C has much less current than Cu. What does this mean in terms of activity when placing the modifier on Cu? What happens if this data is normalized by the ESCA as obtained from sup Fig 18?

Fig3 c, CO₂RR vs CORR is quite surprisingly very different for C₂H₄ ..

Fig3 d,e could have same amount of tickmarks.

There is no discussion of Fig. 3 in the main body text. It seems labels are mixed with Fig 2. Please fix this, parts of the manuscript is very hard to follow when figure references are wrong.

“It is known that the mass transport of CO₂ (local CO₂ concentration) can affect the coverage of *CO₂ (a precursor of *CO), *CO, and *H, which affects the reaction pathways toward ethylene and ethanol in

further” From where is this known? How is mass transport linked to the coverage of intermediates? Coverage is usually given from the binding energies and the chemical potentials – not mass transport properties.

“To circumvent this issue, computational fluid dynamic (CFD) simulations were employed to investigate the mass transport of CO₂ and H₂O in the catalyst layers related to *CO and *H” please consider the usage of *CO and *H throughout the manuscript.

“The coverage of key intermediate *CO can be improved by directly using CO as reactants.” a statement without any explanation. Why is this?

“As another key intermediate, *H derives from the bulk solution, which is affected by H₂O transport.” Please consider the usage of *H (there is no *H in solution).

Reviewer 1:

General Comments R1: *In this work, Lin and coworkers present a mechanistic study on the influence of cathode surface wettability on the CO₂ reduction reaction pathways towards ethanol vs. ethylene. The authors modify the surface of Cu gas diffusion electrodes with alkanethiol self-assembled monolayers with different lengths. With increasing SAM length, the electrodes exhibit an increased hydrophilic behavior. The authors correlate the different surface properties with the different selectivity towards ethanol and ethylene, presenting a mechanistic picture on the role of H₂O, CO_x and H transport and interaction.*

The work is interesting and could be suitable for this journal provided some important points were considered.

Response: We thank the reviewer very much for the comments. We have revised this manuscript carefully and improved relative details.

Specific Comments R1-1: *Could the authors comment on the presence (and role) of Cu²⁺ on bare Cu samples (lacking in the -SH functionalized electrodes). Do these results imply that -SH bonding occurs through previously oxidized Cu sites? On the XPS results, do the authors observe any shift or artifact due to the lower conductivity of -SH samples?*

Response: We thank the reviewer for the useful comment very much. The Cu electrodes were prepared by direct current (DC) sputtering (details in “Methods”) that results in metallic Cu without Cu²⁺. Once the Cu electrode has been prepared, the thiol modification was performed immediately. Even so, the Cu surface would be oxidized (to Cu²⁺) inevitably in ambient during the rapid sample transfer. Accordingly, -SH bonding was likely to occur through previously oxidized Cu sites. However, the presence of Cu²⁺ did not affect the subsequent thiol modification and CO₂RR process (Fig. 1d, e), which is consistent with previous report (*Nat. Mater.*, 2019, 18, 1222–1227). For a better comparison, we have added the XPS data of Ar etched bare Cu sample in Supplementary Fig. 6a. After Ar etching, the bare Cu sample presents Cu 2p peaks at 932.5 eV and 952.4 eV, corresponding to Cu⁺/Cu⁰ species (Supplementary Fig. 6a). The Auger electron spectroscopy (AES) of Cu LMM further indicates that Cu mainly consists of Cu⁰ (918.6 eV) after Ar etching (Supplementary Fig. 6). Moreover, we did not observe any shift or artifact of -SH samples by charge accumulation on XPS result, which is own to the good conductivity of the modified catalysts.

We have added associated discussions on the page 6 in the revised manuscript. Further, the Cu 2p XPS spectra and Auger Cu LMM spectra of bare Cu samples before and after Ar etching have been added in the Supplementary Information (Page 6-7, Supplementary Fig. 5, 6).

“...The Cu-S coordinate bonds were formed via alkanethiolation as illustrated by X-ray photoelectron spectroscopy (XPS) of Cu 2p and Auger electron spectroscopy (AES) of Cu LMM. Before modification, the Cu catalyst consists of Cu⁺/Cu⁰ (932.5/952.4 eV) and Cu²⁺ (934.8/954.6 eV and 943.6/962.7 eV) (Fig. 1d, Supplementary Fig. 5)³⁵. For a better comparison, the surface of the bare Cu sample was analyzed in XPS with Ar etching. After Ar etching, the bare Cu sample presents Cu 2p peaks at 932.5 eV and 952.4 eV, corresponding to Cu⁺/Cu⁰ species (Supplementary Fig. 6a). The AES of Cu LMM further indicates that the bare Cu sample mainly consists of Cu⁰ (918.6 eV) after Ar etching (Supplementary Fig. 6b)^{36,37}...”

Supplementary Fig. 5 | a, Deconvoluted XPS peaks and b, Auger Cu LMM spectrum of Cu sample.

Supplementary Fig. 6 | a, Deconvoluted XPS peaks and b, Auger Cu LMM spectrum of Cu sample after Ar etching. The Cu sample mainly consists of Cu⁰ (918.6 eV) after Ar etching.

Specific Comments R1-2: What would then be the impact once the samples reduce at cathodic potential? This could lead to different reconstruction processes, and difference Cu facets, which might affect local H and CO adsorption and selectivity.

Response: Thanks to the reviewer for the valuable comment. As different Cu facets feature distinctive OH⁻ electrochemical adsorption behaviors, the surface structures of these samples were probed by the OH⁻ electroadsorption technique. The OH⁻ adsorption peaks of Cu, Cu-12C and Cu-12C after reaction are similar, which indicates that the reconstruction of Cu facets at cathodic

potential can be neglected. Further, the structures of Cu-*x*C electrode were also evaluated by XRD, XPS, TEM, CA measurement after 6.5 hours stability test, showing negligible structural variation of as prepared catalyst, which is consistent with the stable selectivity and activity of the catalyst. These results can indicate a relatively stable surficial structure of Cu at cathodic potential.

We have added the characterization of facet exposure of Cu, Cu-12C and Cu-18C before and after reaction in the Supplementary Information (Page 20, Supplementary Fig. 19). The OH⁻ electroadsorption measurements has been added in the revised manuscript (Page 20, Methods).

Supplementary Fig. 19 | Characterization of facet exposure of **a**, Cu, **b**, Cu-12C and **c**, Cu-18C before and after reaction. The OH⁻ adsorption peaks of Cu, Cu-12C and Cu-12C after reaction are similar, which indicates that the reconstruction of Cu facets at cathodic potential can be neglected.

OH⁻ electroadsorption measurements. In-situ OH_{ads} studies were conducted by flowing Ar in the flow cell. Before electrolysis, cyclic voltammetry (20 mV/s) was performed. Then, CO₂ electrolysis was conducted at a constant potential of -1.2 V versus the RHE for 6.5 h by switching the gas feed to CO₂ and flowing the electrolyte. Immediately after electrolysis, the electrolyte (1 M KOH (aq.)) flow rate was stopped to minimize the fluctuation in the voltammogram, and the gas feed was switched to Ar, the electrolyte flow rate was stopped, and then cyclic voltammetry (20 mV/s) was performed. Electrochemical measurements are carried out with a potentiostat (Autolab PGSTAT204, Metrohm).

Specific Comments R1-3: *More details are needed on the SAM incubation. It seems that the formation of the thiol layer proceeds in saturated thiol solutions? (or is a mM concentration fixed). This would lead to uncontrollable coatings, limited by hindrance and surface competition. E.g., a different mols/nm² loading of coated molecules for different number of carbons. While this might*

not affect the ultimate hydrophilic, it could have implications on the surface accessibility of reactants, and other electronic properties.

Response: We thank the reviewer for the kind comment very much. In fact, the surficial modifying of the thiol layer on the copper surface was performed in pure thiol liquid to ensure the saturated adsorption. After this, the electrode was moved to a solution of ethyl acetate to remove the unadherent thiol. We have added more details of the SAM incubation in the experiment section in the revised manuscript (Page 18, Methods).

Preparation of Cu-4C, Cu-7C, Cu-12C, Cu-18C. The Cu electrode was submerged into the 1-dodecanethiol (liquid, 98%, J&K Scientific Ltd.) under Ar and left for 1 min. The formation of the thiol layer (Cu-*x*C) was performed in pure thiol liquid to ensure that thiol molecules reach saturated adsorption on the copper surface. After this, the electrode was moved to a solution of ethyl acetate (99.5%, J&K Scientific Ltd.) for 5 min to remove residual 1-dodecanethiol and then was dried under vacuum at 60 °C, and the resultant electrode was denoted as Cu-12C. The Cu-4C, Cu-7C can be obtained by the same method from 1-butanethiol (97%, J&K Scientific Ltd.) and 1-heptanethiol (98%, J&K Scientific Ltd.), respectively. As for Cu-18C, 1-octadecanethiol (97%, J&K Scientific Ltd.) was first melted under vacuum at 60 °C, and the Cu electrode was submerged into the 1-octadecanethiol under Ar and left for 10min. Then, the electrode was moved to a solution of ethyl acetate at 60 °C for 5 min to remove residual 1-octadecanethiol and was dried under vacuum at 60 °C.

Specific Comments R1-4: At which voltages/pH conditions do the authors expect the SAM to be stable or reduced?

Response: We thank the reviewer for the helpful comment very much. The SAM and Cu catalyst are stable in our experimental condition (1 M KOH (aq.), pH 14, -1.2 V vs. RHE). We have conducted 6.5 hours stability test in 1 M KOH (aq.) (pH 14) at -1.2 V vs. RHE and the durability of the alkanethiol monolayer (SAM) and Cu electrode was evaluated by XRD, XPS, TEM, CA and OH⁻ electroadsorption measurement. The stability test shows a stable reduction current density with a stable product selectivity during durability test (Supplementary Fig. 17). The morphology and exposed facets of Cu catalyst are not significantly changed by the SEM images, XRD profiles and OH⁻ electroadsorption measurement (Supplementary Fig. 18, 19, 20). The XPS analysis also reveal similar Cu/S ratios on catalyst surface before and after the electrolysis (Supplementary Fig. 21). The HRTEM displays the reservation of the alkanethiol layer on the Cu catalyst after reaction, which indicates the high electrochemical stability of hydrophobic layer (Supplementary Fig. 22). Thus, the larger contact angle can be observed after electrolysis. (Supplementary Fig. 23). In this regard, the SAM and Cu catalyst are stable in our experimental condition (1 M KOH (aq.), pH 14, -1.2 V vs. RHE), which is in accordance with previous reports (*Nano Energy*, 2022, 92, 106784).

To make it clear for the readers, we have added associated discussions in the revised manuscript (Page 7).

“...The durability of the alkanethiol monolayer and Cu electrode was evaluated by XRD, XPS, TEM, contact angle measurement and OH⁻ electroadsorption measurement. The 6.5 hours stability test at -1.2 V vs. RHE shows that a stable reduction current density and product selectivity are maintain after alkanethiol modification (Supplementary Fig. 17). According to the SEM images,

XRD profiles and OH^- electroadsorption measurement, the morphology and dominant exposed facets of Cu catalyst are not significantly changed (Supplementary Fig. 18-20). XPS analysis ... In this regard, the exposed facet, morphology and hydrophobicity of the Cu catalyst and the thiol layer after electrolysis are relatively stable under experimental condition (1 M KOH (aq.), pH 14, -1.2 V vs. RHE), which is in accordance with the previous report^{27,34} ...”

Specific Comments R1-5: *The reported JV curves exhibit a clearly different behavior for samples coated with the SAMs. The authors perform most comparative analysis at a fixed potential vs RHE. This would be fair in terms of thermodynamics (the catalysts and reactants would face ~similar barriers), but would neglect important kinetic and dynamic environment properties such as OH coverage, pH etc. The authors are encouraged to perform similar analysis at fixed currents to offer a more complete vision on the proposed mechanisms.*

Response: We thank the reviewer for the kind suggestion. We have added similar analysis at fixed currents (-100 mA cm^{-2}) to obtain a more complete study of our catalyst. With the increasing hydrophobicity, the FE of ethylene increases gradually, while the FE of ethanol increases initially and then decreases gradually, indicating that increased hydrophobicity could promote mass transport during the reduction. The production of ethanol is suppressed under super-hydrophobicity due to the limited H_2O transport.

We have added associated discussions in the Supplementary Information (Page 16, Supplementary Fig. 15).

Supplementary Fig. 15 | Faradaic efficiencies toward ethanol and ethylene of Cu, Cu-12C and Cu-18C at -100 mA cm^{-2} .

With the increasing hydrophobicity, the FE of ethylene increases gradually, while the FE of ethanol increases initially and then decreases gradually, indicating that increased hydrophobicity can promote mass transport during the reduction. The production of ethanol is suppressed under super-hydrophobicity due to the limited H_2O transport.

Specific Comments R1-6: *On a same note, the authors report different ECSA for SAMs with different lengths. This could be the case considering the previous point (e.g., different loading and less active sites that are occupied by SH bonds). Another important aspect is that the samples might exhibit a different in plane and out of plane resistance. This would complicate ECSA analysis based on double layer capacitance (which assumes a similar R). A proper ECSA analysis is also important considering the dynamic surface competition. Detailed EIS analyses or other techniques would help clarify these points.*

Response: We thank the reviewer for the insightful comment. We have performed Electrochemical impedance spectroscopy (EIS) analysis to evaluate the influence of alkanethiols on the conductivity of modified electrodes, which can support the calculation of ECSA analysis (Supplementary Fig. 27, Supplementary Table 4). All the spectra of EIS are characterized by a high-frequency feature which is associated with the charge transport, and a low-frequency arc is associated with the interfacial mass transport. The modeled equivalent circuit of EIS is inserted in Supplementary Fig. 27, where R_s is the electrolyte resistance, R_{ct} is the charge transfer resistance, C is the double-layer capacitance and W is the Warburg impedance. The fitting procedures of all the samples are reported in Supplementary Table 4. The electrolyte resistance (R_s) of the Cu- x C electrodes is not as substantially perturbed as that of the bare Cu electrode. The charge resistance (R_{ct}) and mass transport resistance (W) is increased with alkanethiols modification. The interfacial electrochemical double-layer capacitance (C) from EIS is also obtained, which can quantitatively evaluate the charge accumulation at the interface. The interfacial electrochemical double-layer capacitance (C) gradually decreases with the increasing of alkyl chain length on alkanethiols. Thus, the variation of charge resistance (R_{ct}), mass transport resistance (W) and interfacial electrochemical double-layer capacitance (C) can be ascribed to the strong hydrophobicity induced less contact with the electrolyte after alkanethiol-modification, which is confirmed with the ECSA analysis.

We have added the ECSA analysis in the revised manuscript (Page 8) and Supplementary Information (Page 28, Supplementary Fig. 27; Page 43, Supplementary Table 4).

“...The variation of charge resistance (R_{ct}), mass transport resistance (W) and interfacial electrochemical double-layer capacitance (C) after alkanethiol-modification in Electrochemical impedance spectroscopy (EIS) analysis can be ascribed to the strong hydrophobicity that retarded the contact with the electrolyte after alkanethiol-modification, which is confirmed with the ECSA analysis (Supplementary Fig. 27, Supplementary Table 4)...”

Supplementary Fig. 27 | EIS spectra of the Cu, Cu-12C and Cu-18C electrodes. The EIS is modeled by equivalent circuit inserted in Supplementary Fig. 27, where R_s is the electrolyte resistance, R_{ct} is the charge transfer resistance, C is the double-layer capacitance and W is the Warburg impedance. The results of the fitting procedures for all the samples are reported in Supplementary Table 4.

The electrolyte resistance (R_s) of the Cu- x C electrodes is not as substantially perturbed as that of the bare Cu electrode. The charge resistance (R_{ct}) and mass transport resistance (W) is increased upon alkanethiols modification. The interfacial electrochemical double-layer capacitance (C) from EIS is also obtained, which can quantitatively evaluate the charge accumulation at the interface. The interfacial electrochemical double-layer capacitance (C) gradually decreases with increasing alkyl chain length on alkanethiols. Thus, the variation of charge resistance (R_{ct}), mass transport resistance (W) and interfacial electrochemical double-layer capacitance (C) can be ascribed to the strong hydrophobicity that retarded the contact with the electrolyte after alkanethiol-modification, which is confirmed by the ECSA analysis.

Supplementary Table 4. The fitting parameters of the electrochemical impedance spectra of Cu, Cu-12C and Cu-18C electrodes.

Electrode	R_s ($\Omega \text{ cm}^2$)	R_{ct1} ($\Omega \text{ cm}^2$)	C_1 (F)	R_{ct2} ($\Omega \text{ cm}^2$)	C_2 (F)	W ($\Omega \text{ cm}^2$)
Cu	2.792	0.814	0.014669	30.33	0.0165	21.72
Cu-12C	2.862	1.357	0.010663	36.04	0.01482	36.04
Cu-18C	2.616	7.483	0.000672	79.47	0.0085	274.2

Electrochemical impedance spectroscopy (EIS). EIS measurements were performed in a flow cell at room temperature with Autolab electrochemical workstation. Ni foam and saturated Hg/HgO electrode was used as the counter electrode (CE) and the reference electrode (RE), respectively. The EIS measurements were carried out in 1 M KOH solution at open circuit potential (OCP). The impedance spectra were recorded with an amplitude of 10 mV from 0.01 to 100 kHz. The data obtained from the EIS measurements were fitted by the Zview software (Version 3.1, Scribner Associates, USA).

Specific Comments R1-7: *It is not clear to this reviewer that Fig 5 is representative to the actual catalysts or is adding any value. e.g., the surface of the Cu electrode would be continuous and gas flow and partial CO₂ pressures would be different.*

Response: Thanks to the reviewer for the valuable comment. It is true that there is a gap between our model and the actual catalyst. However, our simplified model is still representative to mimic the effect of wettability on the interfacial contact state and the local CO₂/H₂O ratio. As shown in scanning electron microscopy (SEM) images, Cu nano-islands are uniformly grown on the porous surface of carbon paper with an average size of around 1 μm (Supplementary Fig. 2, 3). Therefore, the surface of the Cu electrode is indeed continuous in macroscale, while the structure of the Cu electrode can be approximated as catalyst islands with pores in microscale. To enable feasible CFD simulation, a simplified model is constructed as shown in Fig. 5a. The squares in the model are used to represent the catalysts, with the various contact angles setting. The gaps between the squares represents the porous surface of carbon paper between the real catalyst islands. The simulation results reveal the formation of continuous liquid layer on the hydrophilic catalyst (Cu) surface and CO₂ needs diffuse through the thin liquid layer to reach the catalyst for reduction, which causes some hinder during gas transport (Fig. 5b). In comparison, the whole surface of high hydrophobicity (CA:156°) catalyst is mostly occupied by gas phase and formed a gas-solid interfaces. The hindrance of gas transport through thin liquid layer is eliminated, which would block H₂O transport from electrolyte, simultaneously (Fig. 5d). In comparison, a gas-liquid-solid interface is formed on the lower hydrophobic catalyst (CA: 131°) with balanced wettability (Fig. 5c, d). The exposure of the three-phase interface could balance the gas and liquid mass transport, resulting in an optimized *CO/*H ratio for ethylene and ethanol conversion. The simulation results also proved that the spreading of gas and electrolyte on catalyst surface can be controlled by wettability, which is consistent with experimental data.

Thus, it is suitable to adopt this CFD model to understand the effect of wettability on the interfacial contact state and the local CO₂/H₂O ratio, which can reveal the mechanism of wettability variation influence on the ethylene and ethanol pathways. Thanks to the reviewers for the helpful suggestion. We have added the explanation of model establishment in the revised manuscript (Page13) to help readers better understand the effectiveness of the model.

“...As shown in SEM images, Cu nano-islands are uniformly grown on the porous surface of carbon paper with an average size of around 1 μm (Supplementary Fig. 2, 3). Therefore, the structure of the Cu electrode can be simplified as catalyst islands with pores. To enable the CFD simulation, a simplified model was constructed (Fig. 5a). The squares in the model are used to represent the catalysts with various contact angles. The gaps between the squares represent the pores between the real catalyst islands...”

Specific Comments R1-8: *The proposed picture in Fig 6 is nice but might be misleading. Have the authors proven that for the EtOH and C₂H₄ reactions CO₂ is coming through the gas phase (and not dissolved in the electrolyte as mentioned in the intro)*

Response: Thanks to the reviewer for the valuable comment. It is true that CO₂ has moderate solubility in aqueous electrolyte, so the reduced CO₂ is ascribed to dissolved CO₂ in the electrolyte in some previous reports. In our case, however, a gas-liquid-solid interface has been built to balance the CO₂ and H₂O transfer, which is essential for effective generate EtOH and C₂H₄ products. In comparison, less EtOH and C₂H₄ products can be detect from the hydrophilic catalyst, although the CO₂ still can be dissolved in electrolyte layer nearby catalyst surface. Moreover, the results of CFD simulation and CLSM indicate that CO₂ come through the gas phase on the hydrophobic-treated electrode, and some previous papers also have reported similar results (*Science*, 2020, 367, 661–666; *Nat. Commun.*, 2021, 12, 136; *Angew. Chem. Int. Ed.*, 2020, 59, 19095-19101;).

Specific Comments R1-9: *Stability should be reported and sufficient to sustain the validity of all the studies presented herein. It is important to see if the SAM functionalization would affect stability.*

Response: Thanks to the reviewer for the useful comment. We have conducted 6.5 hours stability test in 1 M KOH (aq.) at –1.2 V vs. RHE and the durability of the alkanethiol monolayer (SAM) and Cu electrode was evaluated by XRD, XPS, TEM, CA and OH[–] electroadsorption measurement. The stability test shows a stable reduction current density with a stable product selectivity (Supplementary Fig. 17). The morphology and exposed facets of Cu catalyst are not significantly changed by the SEM images, XRD profiles and OH[–] electroadsorption measurement (Supplementary Fig. 18, 19, 20). The XPS analysis also reveal similar Cu/S ratios on catalyst surface before and after the electrolysis (Supplementary Fig. 21). The HRTEM displays the reservation of the alkanethiol layer on the Cu catalyst after reaction, which indicates the high electrochemical stability of hydrophobic layer (Supplementary Fig. 22). Thus, the larger contact angle can be observed after electrolysis. (Supplementary Fig. 23). In this regard, the SAM and Cu catalyst are stable in our experimental condition (1 M KOH (aq.), –1.2 V vs. RHE), which is in accordance with previous reports (*Nano Energy*, 2022, 92, 106784). The results of stability test are shown on the Page 7 in the revised manuscript.

“...The durability of the alkanethiol monolayer and Cu electrode was evaluated by XRD, XPS, TEM, contact angle measurement and OH[–] electroadsorption measurement. The 6.5 hours stability test at –1.2 V vs. RHE shows that a stable reduction current density and product selectivity are maintain after alkanethiol modification (Supplementary Fig. 17). According to the SEM images, XRD profiles and OH[–] electroadsorption measurement, the morphology and dominant exposed facets of Cu catalyst are not significantly changed (Supplementary Fig. 18-20). XPS analysis ... In this regard, the exposed facet, morphology and hydrophobicity of the Cu catalyst and the thiol layer after electrolysis are relatively stable under experimental condition (1 M KOH (aq.), pH 14, –1.2 V vs. RHE), which is in accordance with the previous report^{27,34} ...”

Specific Comments R1-10: *What is the single pass carbon utilization?*

Response: We thank the reviewer for the kind suggestion. The sample calculation of CO₂ conversion is as follow (Joule, 2019, 3, 240–256):

$$\left(\frac{mC}{s\text{ cm}^2}\right)\left(\frac{1C}{1000mC}\right)\left(\frac{mol\ e^-}{96485\ C}\right)\left(\frac{1\ mol\ ethene}{12\ mol\ e^-}\right)\left(\frac{2\ mol\ CO_2}{1\ mol\ ethene}\right)\left(\frac{22.4\ L}{1\ mol\ CO_2}\right)\left(\frac{1000\ ml}{1\ L}\right)\left(\frac{60\ s}{1\ min}\right)(1\ cm^2\ geometric)$$

= mL min⁻¹ CO₂ consumed by device for ethene synthesis

$$100 \times \left(\frac{mL\ min^{-1}\ CO_2\ consumed}{mL\ min^{-1}\ CO_2\ flowed\ in}\right) = \% CO_2\ conversion\ to\ ethene$$

We have added the calculation of the single pass carbon utilization in the revised manuscript (Page 22) and the Supplementary Information (Page 41, Supplementary Table 2).

The calculation of CO₂ conversion. The sample calculation of CO₂ conversion is as follow⁵²:

$$\left(\frac{mC}{s\text{ cm}^2}\right)\left(\frac{1C}{1000mC}\right)\left(\frac{mol\ e^-}{96485\ C}\right)\left(\frac{1\ mol\ ethene}{12\ mol\ e^-}\right)\left(\frac{2\ mol\ CO_2}{1\ mol\ ethene}\right)\left(\frac{22.4\ L}{1\ mol\ CO_2}\right)\left(\frac{1000\ ml}{1\ L}\right)\left(\frac{60\ s}{1\ min}\right)(1\ cm^2\ geometric)$$

= mL min⁻¹ CO₂ consumed by device for ethene synthesis

$$100 \times \left(\frac{mL\ min^{-1}\ CO_2\ consumed}{mL\ min^{-1}\ CO_2\ flowed\ in}\right) = \% CO_2\ conversion\ to\ ethene$$

Supplementary Table 2. Comparison of optimized ethanol and C₂₊ products from various Cu-based catalysts.

Cu-based Catalyst	FE _{C₂₊}	FE _{EiOH}	C ₂₊ partial current density/ mA cm ⁻² (% CO ₂ conversion to C ₂₊)	Ref.
Cu-12C	86.1%	53.7%	103 (1.20%)	This work
Cu-12C	80.3%	47.8%	321 (3.73%)	This work
Cu dendrite	74%	17%	22	Nat. Mater. 2019 ⁵
FeTPP/Cu	80%	41.2%	97	Nat. Catal. 2020 ⁸
Cu-Ag	50%	~20%	160	Joule 2020 ⁹
GB-Cu	73.1%	31.7%	58	J. Am. Chem. Soc. 2020 ¹⁰
Hex-2Cu-O	66.2%	32.5%	182	Nat. Commun. 2022 ¹¹
CSVE-Cu	58.8%	32%	126	Nat. Catal. 2018 ¹²
AgCu	68%	41.4%	170	J. Am. Chem. Soc. 2019 ¹³
ZnO-CuO	41%	32%	80	Angew. Chem., Int. Ed. 2019 ¹⁴
Cu-copolymer	77%	22%	3.6	J. Am. Chem. Soc. 2021 ⁴
Cu-Cu ₂ O	80%	39.2%	5.8	Nat. Commun. 2019 ¹⁵
Cu-N-C	80%	43%	12.9	Angew. Chem., Int. Ed. 2019 ¹⁶

Specific Comments R1-11: What is the thickness of the Cu electrodes? This is not clear in the methods section.

Response: We thank the reviewer for the useful comment very much. We have added cross-sectional SEM images of the Cu electrode. As shown in Supplementary Fig. 3, the area between the two yellow lines represents the catalyst layer. The thickness of the Cu electrode is around 1 μm .

We have added the thickness of the Cu electrodes in the revised manuscript (Page 5) and the Supplementary Information (Page 4, Supplementary Fig. 3).

“...The cross-sectional SEM image also shows that the thickness of the Cu layer is around 1 μm (Supplementary Fig. 3)...”

Supplementary Fig. 3 | Cross-sectional SEM images of the Cu electrode. Yellow lines are added to aid visualization of the catalyst layer. The area between the two yellow lines represents the catalyst layer. The thickness of the Cu layer is around 1 μm .

Specific Comments R1-12: *Elaborate more on the novelty and new findings compared to Nature Materials volume 18, pages 1222–1227 (2019) and other works employing a similar approach.*

Response: The electrochemical conversion of carbon dioxide (CO_2) into value-added multi-carbon (C_{2+}) products with high selectivity is still facing great challenges (*Nature*, 2020, 577, 509; *Science*, 2018, 360, 783; *Science*, 2019, 364, 350). $\ast\text{CO}$ and $\ast\text{H}$ are considered to be the key intermediates during the production of C_{2+} products (*Joule*, 2020, 4, 1104; *Nat. Nanotechnol.*, 2019, 14, 1063). In general, local $\text{CO}_2/\text{H}_2\text{O}$ concentration ratio that relevant with mass transport of CO_2 and H_2O can influence the surface coverage of $\ast\text{CO}$, and $\ast\text{H}$, which affects the reaction pathways toward C_{2+} products (*Joule*, 2020, 4, 1104; *Nat. Commun.*, 2021, 12, 136). Modification of gas diffusion electrode (GDE) is a promising method to improve the transport of reactants and control the coverage of key intermediates (*Science*, 2020, 367, 661; *Nat. Mater.*, 2019, 18, 1222; *Nat. Energy*, 2021, 6, 1026). Although these long side chains containing $-\text{CH}_2-$ (*Nat. Mater.*, 2019, 18, 1222–1227) or $-\text{CF}_2-$ (*Angew. Chem.*, 2020, 132, 160–166; *Nat. Commun.*, 2021, 12, 136) can induce strong hydrophobicity on catalyst surface to improve mass transfer, continuous control of wettability is still meeting great challenges. Moreover, although the C_{2+} selectivity could be

enhanced by using hydrophobic polytetrafluoroethylene (PTFE) layers, it is difficult to identify and optimize the specific effect of hydrophobicity. Thus, the impact of interfacial wettability on the pathways of ethylene and ethanol is rarely understood, thus it is urgently desirable to develop a new approach to continuously tune the *CO/*H ratio by altering the interfacial wettability. This work proposes a direct and efficient interfacial modification protocol to continuously modulate the wettability of the catalyst. Through changing the substituents on alkanethiols, the balance of *CO and *H intermediate on surface can be effectively controlled. Therefore, this manuscript demonstrated the following novelty and new findings: 1) Continuous modulation of a hydrophilic interface to superhydrophobic; 2) Interfacial wettability control for variable *CO/*H ratio; 3) Tunable CO₂/H₂O ratio to optimize *CO/*H ratio for effective production of C₂₊ products. Thus, this work describes the design and realization of controllable equilibrium of *CO and *H hydrophobic via modifying alkanethiols with different alkyl chain lengths to reveal its contribution to ethylene and ethanol pathways.

We have added associated discussions on the page 4 in the revised manuscript.

“...To break the mass transport limitation and improve the selectivity of C₂₊ products, GDE can be modified to improve the transport of reactants and control the coverage of key intermediates^{20,26-32}. Organic molecules (fluorosilane³³, quaternary ammonium salt^{29,30}), polymers (polytetrafluoroethylene^{20,28}), ionic polymers (Nafion^{26,31}) can modulate the local concentrations of CO₂ and H₂O because their backbone chains containing -CH₂- or -CF₂- induce hydrophobicity. Although these long side chains can induce strong hydrophobicity on catalyst surface to improve mass transport effectively, continuous control of wettability is still meeting great challenges^{20,28,33}. Moreover, although the C₂₊ selectivity can be enhanced by using hydrophobic polytetrafluoroethylene (PTFE) layers, it is difficult to identify and optimize the specific effect of hydrophobicity²⁷. Thus, the impact of interfacial wettability on the pathways of products, especially ethylene and ethanol, is rarely understood. Therefore, it is urgently desirable to develop a new approach to continuously tune the *CO/*H ratio by altering the interfacial wettability...”

Specific Comments R1-13: *Language and typos (e.g., ECSA vs ECSA) should be fixed.*

Response: We thank the reviewer for the kind comment very much. We have thoroughly checked and corrected the language and typos in the manuscript (For example, on Page 8).

“...Additionally, the electrochemically active surface area (ECSA) also drops rapidly... Therefore, the lower ECSA and current density with the formation of alkanethiol layer ...”

Reviewer 2:

General Comments R2: *Lin and Wang took an interesting approach to control the surface hydrophobicity of copper catalyst and investigate its impact on the reaction mechanism in the CO₂RR, especially towards ethanol and ethylene formation. They aim to understand how tuning the surface wettability affects the reaction pathway via adjusting the *CO and *H surface coverage. This is an important topic to be discussed in literature. The experiments are mostly designed well, however, the results are not fully supporting the hypothesis in this work. This reviewer believes that the submitted work need to be further improved before being confident to recommend it for publication. The main questions to be answered by the authors are listed below:*

Response: We sincerely appreciate the reviewer's kind comments. We have carefully revised the manuscript based on the comments.

Specific Comments R2-1: *Effect of wettability on *CO coverage is investigated experimentally using ATR-SEIRAS, but there is no direct experimental evidence to show its impact on *H coverage. The HER experiment in Fig. 4 cannot conclusively prove that we have a different *H coverage on the surface. Perhaps Raman would be useful here.*

Response: We thank the reviewer for the useful comment very much. We have tried our best to observe *H coverage directly through Raman, but we still could not observe the peak of *H, because the adsorption of *H on Cu is too weak. As suggested by the reviewer, Raman may be useful for metals with strong binding with *H, such as Pt and Pd. Unfortunately, there is no direct experimental method to observe *H coverage on Cu so far, because the adsorption of *H on Cu is much weaker than that of *CO. At present, the indirect methods of proving *H coverage are as follows: 1) *H coverage indirectly proved through *CO coverage. *H and *CO occupy most of the Cu surface sites, so they are in direct competition with each other for surface sites. When the surface maintains a high *CO coverage, the corresponding *H coverage will decrease (*J. Am. Chem. Soc.*, 2017, 139,15848-15857; *Angew. Chem. Int. Ed.*, 2018,57, 10221-10225; *J. Phys. Chem. Lett.*, 2016, 7, 1471-1477; *Nat. Commun.*, 2021, 12, 5745). 2) *H coverage indirectly proved through water content. Interfacial water content obtained by FT-IR and CLSM (*Nat. Commun.*, 2020, 11, 2038; *J. Am. Chem. Soc.* 2020, 142, 19, 8748–8754).

CO₂ reduction requires water as a H source for hydrocarbon production (*Science*, 2020, 367, 661–666; *Nat. Mater.*, 2019, 18, 1222–1227). Thus, the *H intermediate on surface is converted from the bulk solution (H₂O). In this work, we used the water content to indirectly correlate *H coverage. The HER performance should not be affected by the external gas diffusion, but is determined by H₂O availability and transport. Therefore, the HER performance of Cu catalysts with different wettability can be used to compare the H₂O availability and *H transport, which is inspired by previous work (*Science*, 2020, 367, 661–666). We have revised the statement and added associated discussion in the revised manuscript to avoid misunderstandings. Thanks to the reviewer for the insightful comment. In the future, we are endeavored to develop advanced characterization methods to directly prove *H coverage, which would help us better understand the CO₂RR process.

We have revised the statement and added associated discussion on page 12 in the revised manuscript.

“As another key intermediate, the *H intermediate on surface is converted from the bulk solution, which is affected by H₂O transport. Unfortunately, there is no direct experimental method to observe *H coverage on Cu so far, because the adsorption of *H on Cu is too weak. At present, *H coverage can be investigated indirectly through *CO coverage or H₂O content^{18,46-49}. *H and *CO occupy most of the Cu surface sites, resulting in direct competition between *H and *CO for surface sites. When the surface maintains a high *CO coverage, the corresponding *H coverage will decrease. In order to elucidate the effect of H₂O transport on the ethylene and ethanol reaction pathways, the HER performance of Cu catalysts with different wettability for various electrolytes was compared (Fig. 4a, b). The HER performance should not be affected by the external gas diffusion, but is determined by H₂O availability and transport²⁶. Therefore, stronger hydrophilicity can benefit the HER due to more efficient H₂O transport.”

Specific Comments R2-2: *The role of wettability on *CO and *H coverage is shown (if we accept the discussion for the *H coverage for now), but still, it is not clear why and how these surface coverages affect the reaction pathway towards ethylene and ethanol? That is a key question which is stated in the introduction but not fully answered in this paper. To be more specific, why we have ethanol, and no other alcohols or oxygenates like methanol, propanol or acetate? Why this surface wettability affects ethylene and no other hydrocarbons like methane?*

Response: Thanks to the reviewer for the valuable comment. *H and *CO occupy most of the Cu surface sites, so they are in direct competition with each other for surface sites. When the surface maintains a high *CO coverage, the corresponding *H coverage will decrease (*J. Am. Chem. Soc.*, 2017, 139, 15848-15857; *Angew. Chem. Int. Ed.*, 2018, 57, 10221-10225; *J. Phys. Chem. Lett.*, 2016, 7, 1471-1477; *Nat. Commun.*, 2021, 12, 5745.). Moreover, *CO and *H are considered to be the key intermediates affecting the product selectivity of CO₂RR (*Nat. Catal.*, 2020, 3, 75-82; *Nat. Catal.*, 2019, 2, 1124-1131; *Nat. Commun.*, 2019, 10, 5814; *Joule*, 2020, 4, 1688-1699; *Science*, 2018, 360, 783-787; *ACS Catal.*, 2014, 4, 3742-3748).

Further, we have supplemented the Faradaic efficiency data and analysis of the all products. There is almost no methanol product on copper (*Nat. Energy*, 2019, 4, 732-745), while methane and formate products can be observed. The variation trend of methane Faradaic efficiency is consistent with that of ethylene. The Faradaic efficiencies of formate are relatively higher at low potentials, but lower at high potentials. We also observed that the variation trend of propanol Faradaic efficiency is similar as that of C₂ products, which is because C₂ product is the key intermediate of propanol pathway (*Nat. Commun.*, 2021, 12, 1580.). With the decreasing of CO, the Faradaic efficiency of propanol decreases significantly. However, very few acetates were identified in our experiment, which is due to the different formation mechanism of acetic acid with other C₂ products (ethylene and ethanol). The Faradaic efficiency of acetate in CORR is obviously higher than that of in CO₂RR. The rich *CO coverage on Cu surface in CORR could stabilize ethenone (as a key acetate path intermediate) and inhibit the hydrogen evolution reaction, thus substantially promote acetate formation. A highly alkaline environment (a high pH value) near the electrode-electrolyte interface in CORR can also promote the formation of acetate. Thus, the activity and selectivity of producing acetate were low, and acetate is not the main product in CO₂RR, which also have been reported previously (*Nat. Catal.*, 2022, 5, 251-258; *Nat. Catal.*, 2019, 2, 423-430; *Nat. Catal.*, 2018, 1, 748-755).

We have supplemented the Faradaic efficiency data and analysis of the all products in the Supplementary Information (Page 11, Supplementary Fig. 10). Further, we also have added associated discussions on page 7 and page 12 in revised manuscript.

“...*H and *CO occupy most of the Cu surface sites, resulting in direct competition between *H and *CO for surface sites. When the surface maintains a high *CO coverage, the corresponding *H coverage will decrease...”

“...Moreover, the variation trend of methane Faradaic efficiency is consistent with that of ethylene, while the variation trend of propanol Faradaic efficiency is similar as that of C₂ products (Supplementary Fig. 10)...”

Supplementary Fig. 10 | Product distribution of Cu, Cu-4C, Cu-7C, Cu-12C, and Cu-18C, respectively.

The variation trend of methane Faradaic efficiency is consistent with that of ethylene. The Faradaic efficiencies of formate are relatively higher at low potentials, but lower at high potentials. The variation trend of propanol Faradaic efficiency is similar to that of C₂ products, because C₂ product is the key intermediate of propanol pathway.

Specific Comments R2-3: *In this work it is shown that, the higher *CO coverage is expected when the surface is more hydrophobic. And it's shown that the more hydrophobic surface such as Cu-12C is more selective to produce ethanol compared to ethylene. But then, why in CORR*

experiment we see more ethylene than ethanol compared to the CO₂RR experiment? Should not it be the other way around? I would expect a higher *CO surface coverage in CORR compared to that in CO₂RR, and as it's shown via ATR_SERIRAS test the higher *CO is the consequence of the more hydrophobic surface and logically we should expect more ethanol.

Response: We thank the reviewer for the helpful comment very much. The selectivity of ethanol and ethylene first increases and then decreases (like a volcano) with increasing *CO coverage (*Nat. Catal.*, 2020, 3, 75–82; *Nat. Catal.*, 2019, 2, 1124–1131). As we mention above, the generation of ethanol is in relevance with *CO coverage and *H coverage simultaneously, both of which need to be balanced to achieve high selectivity of ethanol. In our CORR experiment, the direct introduction of CO could lead to excessive *CO coverage, but the *H coverage still be preserved at a certain level as CO₂RR experiment. Thanks for the reviewer's kind suggestion, and we reduce the CO feeding concentration in updated CORR experiment, and more ethanol in CORR can be obtained, which can better support our proposed mechanism.

The updated CORR result and discussion is shown in page 9 and 10 in revised manuscript.

“...Like in the CO₂RR, the direct coupling between two *CO species in the CORR is widely accepted as the major C–C coupling mechanism. Moreover, *CO coverage is determined by the local concentration of CO near catalyst. Thus, the coverage of key intermediate *CO can be improved by directly using CO as reactants. When the Cu catalyst has similar contact angle, the promotion of ethanol production is more obviously than that of ethylene during CORR. In comparison, the similar phenomenon can be observed experiments on CuAg catalyst, which can generate more CO during performing CO₂RR (Supplementary Fig. 32). Therefore, increasing *CO coverage will benefit to ethanol generation.”

Fig. 3 | Effect of controllable wettability on *CO coverage via CO₂ mass transport. a, Top row: schematics display the CO₂RR configuration with (right column) or without (left column) alkanethiol modification. The red dotted box is the simulation area. Second row: comparison of modeled gas availability along the catalyst surface with (right column) or without (left column) alkanethiol modification. Gas availability dramatically increases via alkanethiol-derived hydrophobic environment. **b,** Local CO₂ concentration versus time during the CO₂RR of 100 mA cm⁻² of modeled Cu, Cu-12C and Cu-18C interfacial environments, respectively. A stronger hydrophobicity indicates a faster CO₂ mass transport. **c,** Comparison of ethylene and ethanol Faradaic

efficiencies for CO₂RR versus CORR on Cu, which indicates the improvement of *CO coverage is more favorable for the ethanol pathway than for ethylene. In-situ ATR-SEIRAS spectra of **d**, Cu and (e) Cu-12C, revealing that higher *CO coverage can be maintained on Cu electrodes after hydrophobic treatment. Wherein the stretching band at ~2070 cm⁻¹ corresponds to the stretching band of CO_L on Cu surface.

Specific Comments R2-4: *The potential impact on the reaction pathway is important. By saying potential, we mean the potential on the surface of the catalyst. However, ignoring the iR potential loss cannot give us a realistic picture of what potential exists at the cathode. Therefore, the whole discussion in this manuscript can go under question that perhaps all observations are due to the potential shift, and that's why we see different product distribution. EIS test must be performed to measure the impedance of the system and also to elaborate more on the charge transfer resistance for each catalyst.*

Response: Thanks to the reviewer for the valuable suggestion. We have supplemented the relevance between Faradaic efficiencies of ethanol and various cathode potential after iR correction in Supplementary Fig. 14. Comparing the profiles of Cu, Cu-12C and Cu-18C (Supplementary Fig. 14a and Supplementary Fig. 14b), a similar tendency can be observed before and after iR correction. Therefore, the different product distribution is not due to the potential shift.

Further, we have performed electrochemical impedance spectroscopy (EIS) analysis to evaluate the influence of alkanethiols on the conductivity of modified electrodes, which can support the calculation of ECSA analysis (Supplementary Fig. 27, Supplementary Table 4). All the spectra of EIS are characterized by a high-frequency feature which is associated with the charge transport, and a low-frequency arc associated with the interfacial mass transport. The modeled equivalent circuit of EIS is inserted in Supplementary Fig. 27, where R_s is the electrolyte resistance, R_{ct} is the charge transfer resistance, C is the double-layer capacitance and W is the Warburg impedance. The fitting procedures of all the samples are reported in Supplementary Table 4. The electrolyte resistance (R_s) of the Cu- x C electrodes is not as substantially perturbed as that of the bare Cu electrode. The charge resistance (R_{ct}) and mass transport resistance (W) is increased with alkanethiols modification. The interfacial electrochemical double-layer capacitance (C) from EIS is also obtained, which can quantitatively evaluate the charge accumulation at the interface. The interfacial electrochemical double-layer capacitance (C) gradually decreases with the increasing alkyl chain length on alkanethiols. Thus, the variation of charge resistance (R_{ct}), mass transport resistance (W) and interfacial electrochemical double-layer capacitance (C) can be ascribed to the strong hydrophobicity induced less contact with the electrolyte after alkanethiol-modification.

We have added associated discussions about iR correction in Supplementary Information (Page 15, Supplementary Fig. 14). Moreover, the EIS analysis is added in the revised manuscript (Page 8) and the Supplementary Information (Page 28, Supplementary Fig. 27; Page 43, Supplementary Table 4)

Supplementary Fig. 14 | Faradaic efficiencies of ethanol on Cu, Cu-12C and Cu-18C (a, with iR correction and b, without iR correction).

The relevance between Faradaic efficiencies of ethanol and various cathode potential before and after iR correction is shown in Supplementary Fig. 14. Comparing with the profiles of Cu, Cu-12C and Cu-18 C (Supplementary Fig. 14a and Supplementary Fig. 14b), a similar tendency can be observed before and after iR correction. Therefore, the products distribution and reaction efficiency are related with the hydrophobicity of catalyst.

Supplementary Fig. 27 | EIS spectra of the Cu, Cu-12C and Cu-18C electrodes. The EIS is modeled by equivalent circuit inserted in Supplementary Fig. 27, where R_s is the electrolyte resistance, R_{ct} is the charge transfer resistance, C is the double-layer capacitance and W is the Warburg impedance. The results of the fitting procedures for all the samples are reported in Supplementary Table 4.

The electrolyte resistance (R_s) of the Cu-xC electrodes is not as substantially perturbed as that of the bare Cu electrode. The charge resistance (R_{ct}) and mass transport resistance (W) is increased

upon alkanethiols modification. The interfacial electrochemical double-layer capacitance (C) from EIS is also obtained, which can quantitatively evaluate the charge accumulation at the interface. The interfacial electrochemical double-layer capacitance (C) gradually decreases with increasing alkyl chain length on alkanethiols. Thus, the variation of charge resistance (R_{ct}), mass transport resistance (W) and interfacial electrochemical double-layer capacitance (C) can be ascribed to the strong hydrophobicity that retarded the contact with the electrolyte after alkanethiol-modification, which is confirmed by the ECSA analysis.

Supplementary Table 4. The fitting parameters of the electrochemical impedance spectra of Cu, Cu-12C and Cu-18C electrodes.

Electrode	R_s ($\Omega \text{ cm}^2$)	R_{ct1} ($\Omega \text{ cm}^2$)	C_1 (F)	R_{ct2} ($\Omega \text{ cm}^2$)	C_2 (F)	W ($\Omega \text{ cm}^2$)
Cu	2.792	0.814	0.014669	30.33	0.0165	21.72
Cu-12C	2.862	1.357	0.010663	36.04	0.01482	36.04
Cu-18C	2.616	7.483	0.000672	79.47	0.0085	274.2

Electrochemical impedance spectroscopy (EIS). EIS measurements were performed in a flow cell at room temperature with Autolab electrochemical workstation. Ni foam and saturated Hg/HgO electrode was used as the counter electrode (CE) and the reference electrode (RE), respectively. The EIS measurements were carried out in 1 M KOH solution at open circuit potential (OCP). The impedance spectra were recorded with an amplitude of 10 mV from 0.01 to 100 kHz. The data obtained from the EIS measurements were fitted by the Zview software (Version 3.1, Scribner Associates, USA).

Specific Comments R2-5: *In the introduction, one of the main challenges in the field is discussed as ethylene/ethanol selectivity control, however, throughout the manuscript, ethylene is replaced by C_{2+} products invariantly which makes the story difficult to follow and to draw a clear conclusion.*

Response: Thanks to the reviewer for the valuable suggestion. We have checked the manuscript throughout and make some suitable revisions to avoid the misunderstanding. For example (Page 11 and Page 17):

“The local CO_2 concentration can further influence the coverage of $*CO$, which affects the reaction pathways toward ethylene and ethanol...”

“...Therefore, the reaction pathways of ethylene and ethanol can be tuned by kinetic-controlled $*CO/*H$ ratio through controllable wettability...”

Specific Comments R2-6: *To further support the hypothesis that the surface wettability is the only parameter affecting the selectivity (at least in this study), the authors need to repeat the experiment with a different control sample with a given hydrophobicity. For example, applying*

PTFE on the surface, measuring the contact angle, running the CO₂RR, and monitoring the ethanol/ethylene ratio.

Response: Thanks to the reviewer for the valuable comment. We have repeated the CO₂RR experiment with PTFE modification to measure the effect of wettability on the ethanol/ethylene ratio. As shown in Supplementary Fig. 16, only a small amount of PTFE is needed to induce supra-hydrophobicity on catalyst surface, which leads to great challenges in the continuous control of wettability. Moreover, it is difficult to differentiate the specific effect of wettability on ethanol/ethylene ratio over other factors by introducing PTFE layers. Thus, the impact of interfacial wettability on the pathways of products, especially ethylene and ethanol, is difficultly understood by using PTFE.

The ethanol to ethylene ratio on PTFE modified Cu electrodes has been added in Supplementary Information (Page 17, Supplementary Fig. 16).

Supplementary Fig. 16 | Ethanol to ethylene ratio on PTFE modified Cu electrodes in 1 M KOH (aq.) at -1.2 V versus the RHE with various contact angles.

As shown in Supplementary Fig. 16, only a small amount of PTFE is needed to induce supra-hydrophobicity on catalyst surface, which leads to great challenges in the continuous control of wettability. Moreover, it is difficult to differentiate the specific effect of wettability on ethanol/ethylene ratio over other factors by introducing PTFE layers. Thus, the impact of interfacial wettability on the pathways of products, especially ethylene and ethanol, is difficultly understood by using PTFE.

Specific Comments R2-7: Line 120: nano-features are not visible in Supplementary Fig. 2. The scale bar in SEM images is in micro scale.

Response: Thanks to the reviewer for the kind comment. We have revised this mistake on page 3 in the Supplementary Information (Supplementary Fig. 2).

Supplementary Fig. 2 | SEM images of Cu-12C. Cu nano-islands are uniformly-grown on the carbon paper surface with an average size of around **1 μm**.

Specific Comments R2-8: Fig. 3c: the results for Cu-12C are not shown.

Response: Thanks to the reviewer for the helpful suggestion. We have rewritten the caption of Figure 3c to label the results of Cu instead of Cu-12C on page 9 in revised manuscript.

Fig. 3 | Effect of controllable wettability on $\ast\text{CO}$ coverage via CO_2 mass transport. **a**, Top row: schematics display the CO_2RR configuration with (right column) or without (left column) alkanethiol modification. The red dotted box is the simulation area. Second row: comparison of modeled gas availability along the catalyst surface with (right column) or without (left column) alkanethiol modification. Gas availability dramatically increases via alkanethiol-derived hydrophobic environment. **b**, Local CO_2 concentration versus time during the CO_2RR of 100 mA cm^{-2} of modeled Cu, Cu-12C and Cu-18C interfacial environments, respectively. A stronger the hydrophobicity indicates a faster CO_2 mass transport. **c**, Comparison of ethylene and ethanol Faradaic efficiency

enhancement values for CO₂RR versus CORR on Cu, which indicates the improvement of *CO coverage is more favorable for the ethanol pathway than for ethylene. In-situ ATR-SEIRAS spectra of **d**, Cu and (e) Cu-12C, revealing that higher *CO coverage can be maintained on Cu electrodes after hydrophobic treatment. Wherein the stretching band at ~2070 cm⁻¹ corresponds to the stretching band of CO_L on Cu surface.

Specific Comments R2-9: *There are several typos, grammatical mistakes, and difficult sentences to read, even wrong figure numbers (most Fig. 2's must be Fig. 3's). The manuscript needs to be proofread again before resubmission.*

Response: Thanks to the reviewer for the kind suggestion. We have checked up and revised the resubmitted manuscript. For example, on page 9 and page 10 in revised manuscript.

“... Our data (Fig. 2) also imply that the *CO/*H ratio could be controlled by tuning the local CO₂/H₂O ratio through changing wettability of the catalyst...The models with and without the alkanethiol modification layer were established to quantify dissolved CO₂ in solution (Fig. 3a)...”

“... To investigate the local CO₂ concentration of different interfacial wettability, control samples were prepared with different gas-liquid-solid contact, enabling in situ measurements with fluorescence electrochemical spectroscopy (FES) at 100 mA cm⁻² chronopotentiometry (Fig. 3b)...”

“...The hydrophobic treatment also can promote CO diffusion in CO reduction reaction. Thus, it can be excluded that the limited CO₂ mass transport at the hydrophilic Cu electrode is entirely due to the neutralization of CO₂ with the electrolyte (Fig. 3c, Supplementary Fig. 23) ...”

“...The local CO₂ concentration can further influence the coverage of *CO, which affects the reaction pathways toward C₂₊ products. Thus, in-situ ATR-SEIRAS was employed to further evaluate the impact of interfacial wettability on the adsorption of intermediates (Fig. 3d, e) ...”

Specific Comments R2-10: *Line 129-130: a table of deconvoluted peaks for XPS results could be provided in SI. Also, a survey can be added.*

Response: Thanks to the reviewer for the valuable comment. The deconvoluted peaks for XPS have been added as Supplementary Fig. 5, 6. The Cu catalyst consists of Cu⁺/Cu⁰ (932.5/952.4 eV) and Cu²⁺ (934.8/954.6 eV and 943.6/962.7 eV) (Fig. 1d, Supplementary Fig. 5). For a better comparison, the surface of bare Cu sample was analyzed in XPS with Ar etching. After Ar etching, the bare Cu sample presents Cu 2p peaks at 932.5 eV and 952.4 eV, corresponding to Cu⁺/Cu⁰ species (Supplementary Fig. 6a). However, the binding energy of Cu⁰ (932.6 eV) and Cu⁺ (952.5 eV) in Cu 2p XPS spectra is too close to be identified, so we have conducted Auger electron spectroscopy (AES) of Cu LMM. The Cu 2p XPS spectra and Auger electron spectroscopy (AES) of Cu LMM both indicate that Cu mainly consists of Cu⁰, Cu⁺ and Cu²⁺, while the Cu catalyst after Ar etching mainly consists of Cu⁰.

We have added associated discussions in the revised manuscript (Page 6) and Supplementary Information (Page 6-7, Supplementary Fig. 5,6).

“...The Cu-S coordinate bonds were formed via alkanethiolation as illustrated by X-ray photoelectron spectroscopy (XPS) of Cu 2p and Auger electron spectroscopy (AES) of Cu LMM.

Before modification, the Cu catalyst consists of Cu^+/Cu^0 (932.5/952.4 eV) and Cu^{2+} (934.8/954.6 eV and 943.6/962.7 eV) (Fig. 1d, Supplementary Fig. 5)³⁵. For a better comparison, the surface of the bare Cu sample was analyzed in XPS with Ar etching. After Ar etching, the bare Cu sample presents Cu 2p peaks at 932.5 eV and 952.4 eV, corresponding to Cu^+/Cu^0 species (Supplementary Fig. 6a). The AES of Cu LMM further indicates that the bare Cu sample mainly consists of Cu^0 (918.6 eV) after Ar etching (Supplementary Fig. 6b)^{36,37} ...”

Supplementary Fig. 5 | a, Deconvoluted XPS peaks and b, Auger Cu LMM spectrum of Cu sample.

Supplementary Fig. 6 | a, Deconvoluted XPS peaks and b, Auger Cu LMM spectrum of Cu sample after Ar etching. The Cu sample mainly consists of Cu^0 (918.6 eV) after Ar etching.

Specific Comments R2-11: Line 129-130: For Cu XPS results, did authors etch the surface of the sample? Cu surface can be easily oxidized (to Cu^{2+}) in the atmosphere, and that's why they

observed a little amount of Cu^{2+} for the Cu sample. This reviewer would suggest etching the surface of the sample first, then running XPS for a better comparison.

Response: Thanks to the reviewer for the helpful comment very much. As the reviewer speculated, the Cu surface can be rapidly oxidized to Cu^{2+} in the atmosphere, which introduced a little amount of Cu^{2+} on catalyst surface. To eliminate misunderstanding and compare more accurately, we have etched the surface of the bare Cu sample by Ar before XPS. After Ar etching, the bare Cu sample presents Cu 2p peaks at 932.5 eV and 952.4 eV, corresponding to Cu^+/Cu^0 species (Supplementary Fig. 6a). The AES of Cu LMM further indicates that the bare Cu sample mainly consists of Cu^0 (918.6 eV) after Ar etching (Supplementary Fig. 6b).

We have added XPS and Auger Cu LMM spectrum in the revised manuscript (Page 6) and Supplementary Information (Page 7, Supplementary Fig. 6).

“...The Cu-S coordinate bonds were formed via alkanethiolation as illustrated by X-ray photoelectron spectroscopy (XPS) of Cu 2p and Auger electron spectroscopy (AES) of Cu LMM. Before modification, the Cu catalyst consists of Cu^+/Cu^0 (932.5/952.4 eV) and Cu^{2+} (934.8/954.6 eV and 943.6/962.7 eV) (Fig. 1d, Supplementary Fig. 5)³⁵. For a better comparison, the surface of the bare Cu sample was analyzed in XPS with Ar etching. After Ar etching, the bare Cu sample presents Cu 2p peaks at 932.5 eV and 952.4 eV, corresponding to Cu^+/Cu^0 species (Supplementary Fig. 6a). The AES of Cu LMM further indicates that the bare Cu sample mainly consists of Cu^0 (918.6 eV) after Ar etching (Supplementary Fig. 6b)^{36,37}...”

Supplementary Fig. 6 | a, Deconvoluted XPS peaks and b, Auger Cu LMM spectrum of Cu sample after Ar etching. The Cu sample mainly consists of Cu^0 (918.6 eV) after Ar etching.

Specific Comments R2-12: Line 135-136: It is believed that adding different alkyl chain changes the contact angle. Is the alkyl chain the only parameter to change the contact angle? Or the amount of alkyl chain is also important. For example, if we have two samples: (1) Adding 1 ml of 12C, (2) Adding 5 ml of 12C, do we get the same contact angle? If it has not been done, a control test like that is suggested.

Response: Thanks to the reviewer for the valuable comment. We have conducted the control test according to the suggestions of the reviewer, and we obtained the same contact angle (1 mL 12C vs. 5 mL 12C). In fact, the formation of the thiol layer proceeds in pure thiol liquid to ensure that thiol molecules reach saturated adsorption on the copper surface. After this, the electrode was washed by ethyl acetate to remove excessive thiol. Either 1 mL or 5 mL 12C can reach saturated monolayer adsorption on the Cu surface, resulting in almost the same contact angle.

We have added controlled experiment in the Supplementary Information (Page 8, Supplementary Fig. 7).

Supplementary Fig. 7 | Contact angle of the Cu electrode modified with different amount of 1-dodecanethiol (12C).

The formation of the thiol layer was performed in pure thiol liquid to ensure that thiol molecules reach saturated adsorption on the copper surface. After this, the electrode was washed by ethyl acetate to remove excessive thiol. Either 1 mL or 5 mL 12C can reach saturated monolayer adsorption on the Cu surface, resulting in almost the same contact angle.

Specific Comments R2-13: Fig. 2: A table with exact FEs for all products at different contact angles should be provided in the SI.

Response: Thanks to the reviewer for kind suggestion. We have provided the table with exact FEs for all products at different contact angles in the Supplementary Information (Page 43, Supplementary Table 3).

Supplementary Table 3. The Faradaic efficiencies of all products at -1.2 V vs. RHE of Cu electrodes (Cu-xC) with different contact angles.

Sample	H ₂ /%	CO/%	HCOOH/%	CH ₄ /%	C ₂ H ₄ /%	C ₂ H ₅ OH/%	C ₃ H ₇ OH/%
Cu	14.2	26.1	4.2	0.8	26.3	23.8	4.3

Cu-4C	12.0	18.5	3.2	2.5	26.4	30.8	5.1
Cu-7C	8.5	7.3	3.4	2.4	27.7	44.0	5.4
Cu-12C	6.7	2.7	1.2	3.2	28.0	53.7	4.4
Cu-18C	4.7	7.9	1.1	3.9	35.6	40.3	6.8

Specific Comments R2-14: Fig. 2e is confusing. It is better to have the same range for the y-axis.

Response: Thanks to the reviewer for kind suggestion. We have revised the y-axis in the same range in the revised manuscript (Page 9, Fig 3e).

Fig. 3 | Effect of controllable wettability on *CO coverage via CO_2 mass transport. **a**, Top row: schematics display the CO_2RR configuration with (right column) or without (left column) alkanethiol modification. The red dotted box is the simulation area. Second row: comparison of modeled gas availability along the catalyst surface with (right column) or without (left column) alkanethiol modification. Gas availability dramatically increases via alkanethiol-derived hydrophobic environment. **b**, Local CO_2 concentration versus time during the CO_2RR of 100 mA cm^{-2} of modeled Cu, Cu-12C and Cu-18C interfacial environments, respectively. A stronger the hydrophobicity indicates a faster CO_2 mass transport. **c**, Comparison of ethylene and ethanol Faradaic efficiency enhancement values for CO_2RR versus $CORR$ on Cu, which indicates the improvement of *CO coverage is more favorable for the ethanol pathway than for ethylene. In-situ ATR-SEIRAS spectra of **d**, Cu and (**e**) Cu-12C, revealing that higher *CO coverage can be maintained on Cu electrodes after hydrophobic treatment. Wherein the stretching band at $\sim 2070\text{ cm}^{-1}$ corresponds to the stretching band of CO_L on Cu surface.

Specific Comments R2-15: Line 177: for the stability test, the authors only showed current density. Product distribution is also important and should be mentioned.

Response: We thank the reviewer for valuable suggestion. The reduction current density and product selectivity are kept stable after alkanethiol-modification. We have added product distribution of the stability test in Supplementary Information (Page 18, Supplementary Fig. 17).

Supplementary Fig. 17 | Stability test of Cu-12C for CO₂ electrolysis over 6.5 h (390 min) in 1 M KOH (aq.) at -1.2 V versus the RHE. The reduction current density and product selectivity are kept stable after alkanethiol-modification.

Specific Comments R2-16: *Supplementary Fig. 7: there are some missing products in these figures, including methane, formate, and acetate. Those products should be added to make sure the total FE is around 100%.*

Response: We thank the reviewer for useful suggestion. We have supplemented the Faradaic efficiency of the all products in the Supplementary Information (Page 11, Supplementary Fig. 10).

Supplementary Fig. 10 | Product distribution of Cu, Cu-4C, Cu-7C, Cu-12C, and Cu-18C, respectively.

Specific Comments R2-17: *Supplementary Fig. 10: the caption is incorrect. Cu-xC, not Cu-12C.*

Response: We thank the reviewer for useful suggestion. We have corrected this mistake in the Supplementary Information (Page 14, Supplementary Fig. 13).

Supplementary Fig. 13 | Ethanol-to-ethylene ratio of Cu-xC at various potentials. The ethanol-to-ethylene ratios increase then subsequently drop with the increasing of contact angle at various potentials.

Specific Comments R2-18: *Supplementary Fig. 11: the FEs should be added to show the stability of the system.*

Response: We thank the reviewer for kind suggestion. The reduction current density and product selectivity are kept stable after alkanethiol-modification. We have added product distribution of the stability test in the Supplementary Information (Page 18, Supplementary Fig. 17).

Supplementary Fig. 17 | Stability test of Cu-12C for CO₂ electrolysis over 6.5 h (390 min) in 1 M KOH (aq.) at -1.2 V versus the RHE. The reduction current density and product selectivity are kept stable after alkanethiol-modification.

Specific Comments R2-19: *Supplementary Fig. 19: is the geometric surface area, correct? Almost $8\text{m}^2/\text{g}$?*

Response: Thanks to the reviewer for kind suggestion. The geometric surface area in Supplementary Fig. 26 should be the Brunauer-Emmett-Teller specific surface areas (S_{BET}). It is difficult to obtain enough BET data for samples prepared by sputtering method. Thus, the copper powder with and without alkanethiol treatment were used to verify the influence of alkanethiol modification on Brunauer-Emmett-Teller specific surface area. We have revised this mistake in the Supplementary Information (Page 27, Supplementary Fig. 26).

Supplementary Fig. 26 | The Brunauer-Emmett-Teller specific surface areas (S_{BET}) of Cu electrodes before and after alkanethiol modification. The S_{BET} is almost unchanged.

Specific Comments R2-20: *While using physical vapor deposition method, it is important to mention the thickness as it affects surface facets and other morphologic characteristics. This information is missing.*

Response: We thank the reviewer for the helpful suggestion. We have added cross-sectional SEM images of the Cu electrode. The area between the two yellow lines represents the catalyst layer. The thickness of the Cu layer is around $1\mu\text{m}$.

We have added the thickness of the Cu electrodes in the revised manuscript (Page 5) and the Supplementary Information (Page 4, Supplementary Fig. 3).

“...The cross-sectional SEM image also shows that the thickness of the Cu layer is around $1\mu\text{m}$ (Supplementary Fig. 3)...”

Supplementary Fig. 3 | Cross-sectional SEM images of the Cu electrode. Yellow lines are added to aid visualization of the catalyst layer. The area between the two yellow lines represents the catalyst layer. The thickness of the Cu layer is around 1 μm.

Specific Comments R2-21: *Cite a few papers to support line 219.*

Response: Thanks to reviewer for the kind suggestion. We have cited associated papers to support line 219 on page 9 in revised manuscript.

“It is known that the mass transport of CO₂ (local CO₂ concentration) can affect the coverage of *CO₂ (a precursor of *CO), *CO, and *H, which affects the reaction pathways toward ethylene and ethanol in further^{5,20,42}.”

- 5 Tan, Y. C., Lee, K. B., Song, H. & Oh, J. Modulating Local CO₂ Concentration as a General Strategy for Enhancing C–C Coupling in CO₂ Electroreduction. *Joule* **4**, 1104-1120 (2020).
- 20 Xing, Z., Hu, L., Ripatti, D. S., Hu, X. & Feng, X. Enhancing carbon dioxide gas-diffusion electrolysis by creating a hydrophobic catalyst microenvironment. *Nat. Commun.* **12**, 136 (2021).
- 41 Li, J. *et al.* Constraining CO coverage on copper promotes high-efficiency ethylene electroproduction. *Nat. Catal.* **2**, 1124-1131 (2019).

Specific Comments R2-22: *Add the synthesis details for the CuAg catalyst.*

Response: We thank the reviewer for the kind suggestion. We have added the synthesis details for the CuAg catalyst.

Preparation of CuAg electrode. Ag catalyst (Ag target, 99.999%, Zhongnuo Advanced Material Technology Co., Ltd.) was deposited onto Cu electrode through direct current (DC) magnetron sputtering system. The base pressure was 2.0×10^{-4} Pa, and the flow rate of Ar was set as 20 standard cubic centimeters per minute (sccm). The RF power was 10 W, and the working pressure was 1 Pa. The deposition time was 10 seconds.

Specific Comments R2-23: In addition to the CFD simulations, density functional theory (DFT) computations can leverage the quality of the work, especially by looking into the adsorption energy (or if possible, the reaction pathways) of *CO and *H on different Cu-xC catalysts.

Response: We thank the reviewer for valuable suggestion. The results of density functional theory (DFT) computations indicate that the adsorption energy of *CO and *H on Cu-xC catalysts with different alkyl chain lengths was similar. Further, the adsorption of *H on Cu(111) is much weaker than that of *CO. We have added DFT computations in the revised manuscript (Page 16, 22) and Supplementary Information (Page 38-39, Supplementary Fig. 37, 38).

“...Compared with the classical modulation of active site, the novelty of our method is that the CO₂RR is completely controlled by kinetics (mass transport) through controllable interfacial wettability (Supplementary Fig. 37, 38).”

Supplementary Fig. 37 | Structure model of a, c, Cu(111)-4C and b, d, Cu(111)-12C.

Supplementary Fig. 38 | Adsorption energy of *CO and *H on Cu(111)-4C and Cu(111)-12C.

The adsorption energy of *CO and *H on Cu-xC catalysts with different alkyl chain lengths is similar. Further, the adsorption of *H on Cu(111) is much weaker than that of *CO.

Density Functional Theory (DFT) Methods. The theoretical calculation was conducted by Vienna ab initio simulation package (VASP) with the BEEF-vdW exchange-correlation functional. As for the simulation models, we built five-layer Cu(111)-(4×4) slabs with a Butyl Mercaptan and 1-dodecanethiol molecular to compare the effect of different lengths of thiols on the reaction mechanism. Three bottom layers were fixed while the upper layers were relaxed in these models. The periodic interactions between repeated slabs were minimized by the vacuum space of at least 15 Å. As for the calculation settings, the cut-off energy is 400 eV. The interactions between the atomic cores and electrons were described by the projector augmented wave (PAW) method. All structures were optimized until the force on each atom has been less than 0.02 eV/Å. A (3×3×1) k-point grid were employed for the simulation models.

The free energy of the *CO and *H (ΔG_{ads}) was calculated as follows:

$$\Delta G_{*CO} = G_{*CO} - G_{CO(\text{gas})} - G_{\text{surface}}$$
$$\Delta G_{*H} = G_{*H} - 1/2 G_{H2(\text{gas})} - G_{\text{surface}}$$

Reviewer 3:

General Comments R3: *The study investigates CO₂RR using different modifiers. There are several works currently published on this topic, however I do find parts of this work interesting. But I have multiple comments:*

Response: We appreciate the reviewer very much for the valuable comments. We have carefully revised the manuscript in accordance with the reviewer's comments.

Specific Comments R3-1: *Please clarify what the authors mean when they put asterisk * on their molecules “*CO and *H”. The typical convention is that * means that CO is adsorbed on the surface.*

Response: Thanks to the reviewer for helpful suggestion. * means that CO is adsorbed on the surface. We have clarified * on page 2 and page 3 in our revised manuscript.

“**Abstract:** ...This paper describes the design and realization of controllable equilibrium of *CO and *H (* denotes the adsorbed species on the surface) hydrophobic via modifying alkanethiols with different alkyl chain lengths to reveal its contribution to ethylene and ethanol pathways...”

“... *CO and *H (* denotes the adsorbed species on the surface) are considered to be the key intermediates during producing C₂₊ products^{4,6}. Previously, *CO coverage has been widely accepted as one crucial factor on selective produce ethylene^{4,5}...”

Specific Comments R3-2: *The discussion of, or the framing of, *CO/*H feels a bit out of place. As a GDE does not change the binding energies of *CO or *H, but it rather changes the diffusion or chemical potential at the interface of the species, resulting in an improved performance of a GDE over a normal aqueous test. I believe the authors should be more clear on this point – as now this is mixed together. One example is:*

“*This work describes the design and realization of a strategy that continuously tunes the controllable equilibrium of *CO and *H by altering the interfacial wettability via alkyl thiols with different alkyl chain lengths.*”

Response: Thanks to the reviewer for valuable suggestion very much. We strongly agree with the reviewer that a GDE does not change the binding energies of *CO or *H. We apologize that we did not distinguish thermodynamic and kinetic factors well in the manuscript, which leads to misleading. In fact, catalytic reaction is related to both thermodynamics and kinetics. When the mass transfer rate is fast, the reaction rate is determined by thermodynamics. When the mass transfer is limited, the reaction rate is determined by kinetics. In this work, the catalytic reaction is controlled by kinetics (mass transfer) without changing thermodynamics due to the sluggish mass transfer. Specifically, we strove to increase the reaction rate of one step without strongly modulating the others, by judiciously increasing the coverage of one key intermediate, yet not interfering with the electronic structure (hence binding strength) of Cu. As in previous report, some molecular-metal composite was developed to generates a high concentration of the key early intermediate, yet does not modulate the metallic active sites germane to the crucial C–C coupling

step, which is essentially similar to our work, but through a different method (*Nat. Catal.*, 2020, 3, 75–82).

An independent means of controlling CO coverage is by changing the CO partial pressure or local CO concentration ([CO]), where the local CO concentration correlates with p_{CO} by Henry's law. At equilibrium, the surface coverage of CO (θ_{CO}) is directly proportional to the local CO partial pressure, as given by the following equation (*Nat. Catal.*, 2019, 22, 1124–1131):

$$\theta_{CO} = \theta \times [CO] \times e^{-\frac{E_{CO}}{RT}}$$

Where θ is the coverage of free surface sites, E_{CO} is the CO adsorption energy on the surface, R is the ideal gas constant and T is the temperature. The coverage of intermediate (θ_{CO}) is a function of CO adsorption energy (E_{CO}) and local CO concentration ([CO]). To vary the CO coverage, changing the local concentration of CO at catalysts layer rather than CO adsorption energy is therefore a promising approach. Compared with the classical modulation of active site, the novelty of our method is that the CO₂RR is completely controlled by kinetics (mass transport) through controllable interfacial wettability, which has not been reported so far.

Our method is not as direct as changing the adsorption energy. The classical design of active site is still very important, but our work provides a new approach on catalyst design based on the mass transport for future industrial application.

We have added associated explanations on page 4 and page 16 in revised manuscript.

“...Further, the reaction pathways on catalyst surfaces with different wettability can be understood (Fig. 6c). The surface coverage of adsorbed *CO_x (θ_{CO_x} , x=1 or 2) on the catalyst is proportional to the local concentration of CO_x ([CO_x]), which is given as Equation 1^{5,42}:

$$\theta_{CO_x} = \theta \times [CO_x] \times e^{-\frac{E_{CO_x}}{RT}} \quad (1)$$

where θ is the coverage of free surface sites, E_{CO_x} is the CO_x adsorption energy on the surface, R is the ideal gas constant and T is the temperature. The coverage of intermediate (θ_{CO_x}) is a function of CO adsorption energy (E_{CO_x}) and local CO concentration ([CO_x]). To vary the *CO coverage, changing the local concentration of CO at catalysts layer rather than *CO adsorption energy is therefore a promising approach. Compared with the classical modulation of active site, the novelty of our method is that the CO₂RR is completely controlled by kinetics (mass transport) through controllable interfacial wettability (Supplementary Fig. 37, 38)...”

“This work describes the design and realization of tunable interfacial wettability through using different alkyl thiols. Then, the local concentration of CO₂ and H₂O can be modulated by changing CO₂ and H₂O transport through different interface wettability, which can achieve an optimized equilibrium of kinetic-controlled *CO and *H in a controllable manner...”

Specific Comments R3-3: *There are several sentences/statements with misleading references which needs to be corrected. Particularly in the introduction, I give some examples here:*

“*CO and *H are considered to be the key intermediates during producing C₂₊ products”
Indeed *CO and *H are key intermediates for a predictive scheme for CO₂RR as first shown by Bagger et al. <https://doi.org/10.1002/cphc.201700736>

“As the only metal with a negative adsorption energy for *CO but a positive adsorption energy for *H, copper (Cu) presents a unique ability to produce C₂₊ products^{9,10}. However, multiple products have been detected on Cu surfaces resulting in poor product selectivity for Cu¹¹.”

Please refer to the original and first publications by Bagger et al. <https://doi.org/10.1002/cphc.201700736> and <https://doi.org/10.1021/acscatal.9b01899>

“Previous investigations have found that the coverage of *CO and *H on Cu surface plays a critical role on determining selectivity of C₂₊ products¹².” Ref 12: *Energy Environ. Sci.* 5, 7050–7059 (2012). ref 12 does not discuss *CO and *H on the Cu plays a critical role.

Response: We thank the reviewer for helpful suggestion. Bagger et al. have first proposed four non-coupled binding energies of intermediates (H*, COOH*, CO*, and CH₃O*) as descriptors, for predicting the product distribution in CO₂ electroreduction. This work helps us understand what surface properties determine the main product of CO₂ reduction (A. Bagger et al., *ChemPhysChem* 2017, 18, 3266–3273). Bagger et al. also originally found that two descriptors, ΔE_{CO^*} and ΔE_{H^*} , could explain the further reduction to products beyond CO* (A. Bagger et al., *Catal. Today* 2017, 288, 74–78). Further, they proposed that the Cu catalyst is unique for CO₂ reduction reactions, as compared with other metals, because of its ability to produce a wide range of hydrocarbon and oxygenated products (A. Bagger et al., *ACS Catal.*, 2019, 9, 7894–7899). Cu is the only metal catalyst to give products beyond CO, which is because Cu binds CO* while not having H_{upd} (A. Bagger et al., *ChemPhysChem* 2017, 18, 3266–3273). Moreover, they also proposed that the analysis for the ratio of diffusion between CO₂ and H₂O helping to shed light on a likely mechanism behind the changes in selectivity for functionalized Cu surfaces (A. Bagger et al., *ACS Catal.*, 2022, 12, 15737–15749). Importantly, they further expanded these studies to bond the C atom of CO₂ with other valuable heteroatoms (e.g., N), which is an alternative strategy to produce value-added products and highly beneficial for expanding the application of CO₂RR (A. Bagger et al., *ACS Catal.*, 2023, 13, 1926–1933). A series of innovative work by Bagger et al. has given us a deeper understanding of the critical role of key intermediates on the Cu for C₂₊ product production.

We have corrected the misleading references on page 3 in revised manuscript.

“...*CO (* denotes the adsorbed species on the surface) and *H are considered to be the key intermediates during producing C₂₊ products⁴...”

“...As the only metal with a negative adsorption energy for *CO but a positive adsorption energy for *H, copper (Cu) presents a unique ability to produce C₂₊ products^{4,9}. However, multiple products have been detected on Cu surfaces resulting in poor product selectivity for Cu^{9,10}...”

“...Previous investigations have found that the coverage of *CO and *H on Cu surface plays a critical role on determining selectivity of C₂₊ products^{4,11}...”

4 Bagger, A., Ju, W., Varela, A. S., Strasser, P. & Rossmeisl, J. Electrochemical CO₂ Reduction: A Classification Problem. *Chemphyschem* **18**, 3266-3273 (2017).

9 Bagger, A., Ju, W., Varela, A. S., Strasser, P. & Rossmeisl, J. Electrochemical CO₂ Reduction: Classifying Cu Facets. *ACS Catal.* **9**, 7894-7899 (2019).

- 10 Christensen, O. *et al.* Can the CO₂ Reduction Reaction Be Improved on Cu: Selectivity and Intrinsic Activity of Functionalized Cu Surfaces. *ACS Catal.* **12**, 15737-15749 (2022).
- 11 Bagger, A., Ju, W., Varela, A. S., Strasser, P. & Rossmeisl, J. Single site porphyrine-like structures advantages over metals for selective electrochemical CO₂ reduction. *Catalysis Today* **288**, 74-78 (2017).

Specific Comments R3-4: *CO₂RR experiments carried out in 1M KOH. Fig 2 “e, Faradaic efficiencies of ethanol and ethylene on Cu electrodes in 1 M KOH (aq.) at -1.2 V versus the RHE with various contact angles (without iR correction).” When KOH is saturated with CO₂, it is not KOH anymore, but rather (bi)carbonate solution. Please clarify.*

Response: Thanks to the reviewer for valuable comment. As the reviewer commented, the electrolyte KOH will be neutralized by CO₂ after a long time of reaction, resulting in the kinetic degradation. In the flow cell, the flowing of electrolyte can refresh the local pH on electrode surface. Moreover, the electrolyte needs to be renewed after a long operation. For our work, CO₂ comes through the gas phase after the hydrophobic treatment. Compared with the effect of mass transport, the effect of electrolyte acidification is relatively small, which is consistent with previous reports (*Science*, 2020, 367, 661–666; *Nat. Commun.*, 2021, 12, 136). Therefore, mass transport is still the main influencing factor in our work, and the acidification of the electrolyte does not affect the conclusion of our work.

Specific Comments R3-5: *Fig2 f, clearly the Cu-12C and Cu-18C has much less current than Cu. What does this mean in terms of activity when placing the modifier on Cu? What happens if this data is normalized by the ECSA as obtained from sup Fig 18?*

Response: Thanks to the reviewer for helpful comment. The currents have been normalized by the ECSA in the revised manuscript (Supplementary Table 5). The lower ECSA and current density after modified alkanethiol layer can be ascribe to the reduced contact with the aqueous electrolyte (Fig. 2f, Supplementary Fig. 18, 19). However, Cu-12C (93.9 mA·cm⁻²) has the largest normalized current, followed by Cu (68.2 mA·cm⁻²) and Cu-18C (64.5 mA·cm⁻²). Therefore, Cu-12C presents a higher C₂₊ Faradaic efficiency, which is not due to the larger ECSA, but due to the higher mass transport efficiency.

Supplementary Table 5. The normalized current density of Cu, Cu-12C and Cu-18C at -1.2 V based on ECSA.

Sample	$j(\text{mA}\cdot\text{cm}^{-2})$	ECSA (cm ²)	FE _{C₂₊} (%)	The normalized current density (mA·cm ⁻²)
Cu	164.9	1.34	55.4	68.2
Cu-12C	120.0	1.10	86.1	93.9
Cu-18C	42.9	0.55	82.7	64.5

As shown in Supplementary Table 5, stronger hydrophobicity of electrode leads to smaller ECSA. However, Cu-12C (93.9 mA·cm⁻²) has the largest normalized current, followed by Cu (68.2

mA·cm⁻²) and Cu-18C (64.5 mA·cm⁻²). Therefore, Cu-12C presents the higher C₂₊ Faradaic efficiency, which is not due to the larger ECSA, but due to the higher mass transport efficiency.

Specific Comments R3-6: Fig3 c, CO₂RR vs CORR is quite surprisingly very different for C₂H₄.

Response: Thanks to the reviewer for the helpful comment very much. The selectivity of ethanol and ethylene first increases and then decreases (like a volcano) with increasing *CO coverage (*Nat. Catal.*, 2020, 3, 75–82; *Nat. Catal.*, 2019, 2, 1124–1131). As we mention above, the generation of ethanol is in relevance with *CO coverage and *H coverage simultaneously, both of which need to be balanced to achieve high selectivity of ethanol. In our CORR experiment, the direct introduction of CO could lead to excessive *CO coverage, but the *H coverage still be preserved at a certain level as CO₂RR experiment. Thanks for the reviewer's kind suggestion, and we reduce the CO feeding concentration in updated CORR experiment, and more ethanol in CORR can be obtained, which can better support our proposed mechanism.

The updated CORR result and discussion is shown in page 9 and 10 in revised manuscript.

“...Like in the CO₂RR, the direct coupling between two *CO species in the CORR is widely accepted as the major C–C coupling mechanism. Moreover, *CO coverage is determined by the local concentration of CO near catalyst. Thus, the coverage of key intermediate *CO can be improved by directly using CO as reactants. When the Cu catalyst has similar contact angle, the promotion of ethanol production is more obviously than that of ethylene during CORR. In comparison, the similar phenomenon can be observed experiments on CuAg catalyst, which can generate more CO during performing CO₂RR (Supplementary Fig. 32). Therefore, increasing *CO coverage will benefit to ethanol generation...”

Fig. 3 | Effect of controllable wettability on *CO coverage via CO₂ mass transport. a, Top row: schematics display the CO₂RR configuration with (right column) or without (left column) alkanethiol modification. The red dotted box is the simulation area. Second row: comparison of modeled gas availability along the catalyst surface with (right column) or without (left column) alkanethiol modification. Gas availability dramatically increases via alkanethiol-derived hydrophobic environment. **b**, Local CO₂ concentration versus time during the CO₂RR of

100 mA cm⁻² of modeled Cu, Cu-12C and Cu-18C interfacial environments, respectively. A stronger the hydrophobicity indicates a faster CO₂ mass transport. **c, Comparison of ethylene and ethanol Faradaic efficiency enhancement values for CO₂RR versus CORR on Cu**, which indicates the improvement of *CO coverage is more favorable for the ethanol pathway than for ethylene. In-situ ATR-SEIRAS spectra of **d**, Cu and (e) Cu-12C, revealing that higher *CO coverage can be maintained on Cu electrodes after hydrophobic treatment. Wherein the stretching band at ~2070 cm⁻¹ corresponds to the stretching band of CO_L on Cu surface.

Specific Comments R3-7: *Fig3 d,e could have same amount of tickmarks.*

Response: Thanks to the reviewer for kind suggestion. We have revised the y-axis with the same amount of tickmarks in Fig 3e.

Specific Comments R3-8: *There is no discussion of Fig. 3 in the main body text. It seems labels are mixed with Fig 2. Please fix this, parts of the manuscript is very hard to follow when figure references are wrong.*

Response: Thanks to the reviewer for the kind comment. We have revised these mistakes on page 9-10 in the revised manuscript.

“... Our data (Fig. 2) also imply that the *CO/*H ratio could be controlled by tuning the local CO₂/H₂O ratio through changing wettability of the catalyst...The models with and without the alkanethiol modification layer were established to quantify dissolved CO₂ in solution (Fig. 3a)...”

“...To investigate the local CO₂ concentration of different interfacial wettability, control samples were prepared with different gas-liquid-solid contact, enabling in situ measurements with fluorescence electrochemical spectroscopy (FES) at 100 mA cm⁻² chronopotentiometry (Fig. 3b)...”

“...The hydrophobic treatment also can promote CO diffusion in CO reduction reaction. Thus, it can be excluded that the limited CO₂ mass transport at the hydrophilic Cu electrode is entirely due to the neutralization of CO₂ with the electrolyte (Fig. 3c, Supplementary Fig. 23)...”

“...The local CO₂ concentration can further influence the coverage of *CO, which affects the reaction pathways toward C₂₊ products. Thus, in-situ ATR-SEIRAS was employed to further evaluate the impact of interfacial wettability on the adsorption of intermediates (Fig. 3d, e)...”

Specific Comments R3-9: *“It is known that the mass transport of CO₂ (local CO₂ concentration) can affect the coverage of *CO₂ (a precursor of *CO), *CO, and *H, which affects the reaction pathways toward ethylene and ethanol in further” From where is this known? How is mass transport linked to the coverage of intermediates? Coverage is usually given from the binding energies and the chemical potentials – not mass transport properties.*

Response: We thank the reviewer for insightful comment. We apologize that we did not cite enough papers to support this statement, as well as distinguish thermodynamic and kinetic factors well in the manuscript, which led to misleading. The relevance between local CO₂ concentration

([CO]) and the surface coverage of CO (θ_{CO}) is shown as equation (*Nat. Catal.*, 2019, 22, 1124–1131; *Joule*, 2020, 4, 1–17):

$$\theta_{CO} = \theta \times [CO] \times e^{-\frac{E_{CO}}{RT}}$$

where θ is the coverage of free surface sites, E_{CO} is the CO adsorption energy on the surface, R is the ideal gas constant and T is the temperature. The coverage of intermediate (θ_{CO}) is a function of CO adsorption energy (E_{CO}) and local CO concentration ([CO]). A promising means of controlling CO coverage is by the CO partial pressure (p_{CO}) or local CO concentration ([CO]), where the local CO concentration correlates with p_{CO} ([CO]) by Henry's law. In this work, the catalytic reaction is controlled by kinetics (mass transport) without changing thermodynamics. Specifically, we strove to increase the reaction rate of one step without strongly modulating the others, by judiciously increasing the coverage of one key intermediate, yet not interfering with the electronic structure (hence binding strength) of Cu. Therefore, catalytic reaction is not only related to thermodynamics, but also to kinetics.

We have cited related papers to support this statement and added associated analysis on page 9 and page 16 in the revised manuscript.

“It is known that the mass transport of CO₂ (local CO₂ concentration) can affect the coverage of *CO₂ (a precursor of *CO), *CO, and *H, which affects the reaction pathways toward ethylene and ethanol in further^{5,20,42}.”

- 5 Tan, Y. C., Lee, K. B., Song, H. & Oh, J. Modulating Local CO₂ Concentration as a General Strategy for Enhancing C–C Coupling in CO₂ Electroreduction. *Joule* **4**, 1104–1120 (2020).
- 20 Xing, Z., Hu, L., Ripatti, D. S., Hu, X. & Feng, X. Enhancing carbon dioxide gas-diffusion electrolysis by creating a hydrophobic catalyst microenvironment. *Nat. Commun.* **12**, 136 (2021).
- 41 Li, J. *et al.* Constraining CO coverage on copper promotes high-efficiency ethylene electroproduction. *Nat. Catal.* **2**, 1124–1131 (2019).

“...Further, the reaction pathways on catalyst surfaces with different wettability can be understood (Fig. 6c). The surface coverage of adsorbed *CO_x (θ_{CO_x} , x=1 or 2) on the catalyst is proportional to the local concentration of CO_x ([CO_x]), which is given as Equation 1^{5,42}:

$$\theta_{CO_x} = \theta \times [CO_x] \times e^{-\frac{E_{CO_x}}{RT}} \quad (1)$$

where θ is the coverage of free surface sites, E_{CO_x} is the CO_x adsorption energy on the surface, R is the ideal gas constant and T is the temperature. The coverage of intermediate (θ_{CO_x}) is a function of CO adsorption energy (E_{CO_x}) and local CO concentration ([CO_x]). To vary the *CO coverage, changing the local concentration of CO at catalysts layer rather than *CO adsorption energy is therefore a promising approach. Compared with the classical modulation of active site, the novelty of our method is that the CO₂RR is completely controlled by kinetics (mass transport) through controllable interfacial wettability (Supplementary Fig. 37, 38)...”

Specific Comments R3-10: “To circumvent this issue, computational fluid dynamic (CFD) simulations were employed to investigate the mass transport of CO₂ and H₂O in the catalyst layers related to *CO and *H” please consider the usage of *CO and *H throughout the manuscript.

Response: Thanks to the reviewer for valuable comment. We are sorry that did not explain how mass transport (local CO₂ concentration) is linked to the coverage of intermediates, which confuses readers. Further, we did not distinguish thermodynamic and kinetic factors well in the manuscript. As mentioned above, the coverage of intermediate is a function of CO adsorption energy and local CO concentration. The surface coverage of adsorbed *CO on the catalyst is proportional to the local concentration of CO₂ (*Nat. Catal.*, 2019, 22, 1124–1131; *Joule*, 2020, 4, 1–17). Thus, catalytic reaction is not only related to thermodynamics, but also to kinetics. To avoid misleading, we have added detailed explanations of the relevance between local CO₂ concentration and the coverage of intermediates on page 16 in the revised manuscript. Further, we have carefully considered and revised the usage of *CO and *H throughout the manuscript. Specifically, we added “kinetic-controlled” in front of *CO or *H to distinguish thermodynamic and kinetic factors, and added “local CO₂ concentration” or “mass transport of CO₂ and H₂O” to link with *CO and *H. For example:

“...To circumvent this issue, computational fluid dynamic (CFD) simulations were employed to investigate the mass transport of CO₂ and H₂O in the catalyst layers related to kinetic-controlled *CO and *H.”

“...Further, the reaction pathways on catalyst surfaces with different wettability can be understood (Fig. 6c). The surface coverage of adsorbed *CO_x (θ_{CO_x} , x=1 or 2) on the catalyst is proportional to the local concentration of CO_x ([CO_x]), which is given as Equation 1^{5,42}:

$$\theta_{CO_x} = \theta \times [CO_x] \times e^{-\frac{E_{CO_x}}{RT}} \quad (1)$$

where θ is the coverage of free surface sites, E_{CO_x} is the CO_x adsorption energy on the surface, R is the ideal gas constant and T is the temperature. The coverage of intermediate (θ_{CO_x}) is a function of CO adsorption energy (E_{CO_x}) and local CO concentration ([CO_x]). To vary the *CO coverage, changing the local concentration of CO at catalysts layer rather than *CO adsorption energy is therefore a promising approach. Compared with the classical modulation of active site, the novelty of our method is that the CO₂RR is completely controlled by kinetics (mass transport) through controllable interfacial wettability (Supplementary Fig. 37, 38)...”

Specific Comments R3-11: “The coverage of key intermediate *CO can be improved by directly using CO as reactants.” a statement without any explanation. Why is this?

Response: Thanks to the reviewer for kind comment. Like in the CO₂RR, the direct coupling between two *CO species in the CORR is widely accepted as the major C–C coupling mechanism in most studies to date (*ACS Catal.*, 2018, 8, 7445–7454; *Chem. Soc. Rev.*, 2021, 50, 12897–12914). Moreover, since θ_{*CO} is determined by the local concentration of CO near catalyst, the yield of C₂₊ products on electrode can be enhanced by increasing local CO concentration (*Nat. Catal.*, 2019, 22, 1124–1131). In another paper, one catalyst selectively converts CO₂ to CO could

provide an in-situ source of CO to enhances θ_{*CO} , and the other Cu-containing catalyst performs C–C coupling (*Nat. Catal.*, 2022, 5, 202–211). Additionally, they adopt CO instead of CO₂ as the feedstock to enhance surface-adsorbed CO species for preparing C₂₊ production, and adsorbed CO species also be considered as intermediate for C–C coupling (*Nat. Catal.*, 2019, 2, 251–258). Therefore, the coverage of key intermediate *CO can be improved by using CO directly as reactants.

We have added associated explanations on page 10 and 16 in the revised manuscript to help readers better understand this statement.

“...Like in the CO₂RR, the direct coupling between two *CO species in the CORR is widely accepted as the major C–C coupling mechanism. Moreover, *CO coverage is determined by the local concentration of CO near catalyst. Thus, the coverage of key intermediate *CO can be improved by directly using CO as reactants...”

“...Further, the reaction pathways on catalyst surfaces with different wettability can be understood (Fig. 6c). The surface coverage of adsorbed *CO_x (θ_{CO_x} , x=1 or 2) on the catalyst is proportional to the local concentration of CO_x ([CO_x]), which is given as Equation 1^{5,42}:

$$\theta_{CO_x} = \theta \times [CO_x] \times e^{-\frac{E_{CO_x}}{RT}} \quad (1)$$

where θ is the coverage of free surface sites, E_{CO_x} is the CO_x adsorption energy on the surface, R is the ideal gas constant and T is the temperature. The coverage of intermediate (θ_{CO_x}) is a function of CO adsorption energy (E_{CO_x}) and local CO concentration ([CO_x)]...”

Specific Comments R3-12: “As another key intermediate, *H derives from the bulk solution, which is affected by H₂O transport.” Please consider the usage of *H (there is no *H in solution).

Response: Thanks to the reviewer for helpful comment. We strongly agree with the reviewer that there is no *H in solution. Further, CO₂ reduction requires water as a H source for hydrocarbon production (*Science*, 2020, 367, 661–666; *Nat. Mater.*, 2019, 18, 1222–1227). Thus, the *H intermediate on surface is converted from the bulk solution. We have reconsidered and revised the usage of *H on page 12 in the revised manuscript.

“As another key intermediate, the *H intermediate on surface is converted from the bulk solution, which is affected by interfacial H₂O transport...”

REVIEWER COMMENTS

Reviewer #1 (Remarks to the Author):

The authors have addressed a substantial number of comments of the previous review. The comments regarding the last figures (R1-7 and R1-8, corresponding to Fig 5 and 6) have not been implemented in full (especially with concerns with the nano-islands) and might represent a highly oversimplified of the electrode. Since they are not crucial to underpin the results (which already correlate contact angle with performance), they could be combined or split into the main MS and SI to improve readability.

Reviewer #2 (Remarks to the Author):

Authors demonstrated a great effort towards responding to the questions. Most of the comments are addressed, however, in a few cases the responses are not yet satisfactory, detailed below:

R2-1: Although nothing is observed using Raman, the question still remains there, and the reviewer believes that the study lacks a direct evidence showing *H coverage.

R2-2: The response is not convincing. This reviewer is not looking for a conclusion based on experimental observation, rather looks for a link between the hypothesis of surface *H/*CO coverage and product distribution. The question is why altering *H/*CO surface coverage, only affects ethylene and ethanol formation, knowing that in experiment we only observe these products. The question is “why” and not “what” we observe.

Reviewer #3 (Remarks to the Author):

I have no further comments.

Reviewer 1:

General Comments R1: The authors have addressed a substantial number of comments of the previous review.

Response: We sincerely appreciate the reviewer's kind comments. We have carefully revised the manuscript based on the comments.

Specific Comments R1-1: The comments regarding the last figures (R1-7 and R1-8, corresponding to Fig 5 and 6) have not been implemented in full (especially with concerns with the nano-islands) and might represent a highly oversimplified of the electrode. Since they are not crucial to underpin the results (which already correlate contact angle with performance), they could be combined or split into the main MS and SI to improve readability.

Response: We thank the reviewer for the useful comment very much. We have realized that the placement of CFD results is not very suitable. Thanks to the reviewer for the useful suggestion, we have split Fig 5 into the main revised manuscript (combined with Fig. 4, page 29) and Supplementary Information (Supplementary Fig. 37, page 38) to improve readability.

Fig. 4 | Effect of controllable wettability on *H coverage via H₂O mass transport. **a**, LSV curves of H₂ evolution on the Cu electrodes with different contact angles (CAs) in 0.5 M H₂SO₄ electrolyte and **b**, 1M KOH electrolyte ($v = 50 \text{ mV s}^{-1}$). The hydrophilic Cu electrode (CA: 43°, blue line) exhibits optimal hydrogen evolution activity, suggesting the efficient H₂O transport in hydrophilic environment. **c**, Schematic illustration of available H₂O concentration at reaction interface of Cu, Cu-12C and Cu-18C, respectively. The catalyst layer

in the hydrophilic (Cu) and super-hydrophobic state (Cu-18C) are dominated by the liquid-solid interfaces and the gas-solid interface, respectively. The catalyst layer with balanced wettability (Cu-12C) is occupied by gas-liquid-solid interface. **d-f**, Cross-sectional fluorescence images (scale bar: 20 μm) and corresponding z axis fluorescence intensity line scans of labelled regions (white arrows) of Cu, Cu-12C and Cu-18C, respectively. The decay distance of the fluorescence intensity increases with the improvement of the hydrophobicity of the catalyst layer, indicating the smaller available H_2O concentration gradient. **g**, Schematic of the CFD simulation. The red and blue parts in the schematic represent gas and liquid, respectively. The squares represent catalysts. The gaps between the squares represent the pores between the real catalyst islands. The different wettability of the catalyst layer will lead to difference in the interfacial contact state and the local $\text{CO}_2/\text{H}_2\text{O}$ ratio. CA: contact angle. **h-j**, Comparison of the modeled gas-liquid mass transport with different interfacial wettability, in which the red color in the simulation result represents an CO_2 volume fraction of 100% (electrolyte 0%), while the blue color represents an CO_2 volume fraction of 0% (electrolyte 100%). The $\text{CO}_2/\text{H}_2\text{O}$ ratio at the reaction interface increases with the enhance of hydrophobicity.

Supplementary Fig. 38 | Schematics of interface structure of Cu (CA: 43°), Cu-12C (CA: 131°), and Cu-18C (CA: 156°) corresponding to the CFD simulation results.

Reviewer 2:

General Comments R2: *Authors demonstrated a great effort towards responding to the questions. Most of the comments are addressed, however, in a few cases the responses are not yet satisfactory, detailed below:*

Response: We appreciate the reviewer very much for the positive evaluation of our work. We have carefully revised the manuscript in accordance with the reviewer's comments. The reviewer's comments are extremely helpful and have made our work more critical.

Specific Comments R2-1: *Although nothing is observed using Raman, the question still remains there, and the reviewer believes that the study lacks a direct evidence showing *H coverage.*

Response: We thank the reviewer for the valuable comment very much. We apologize that we did not explain speculative reasons in the manuscript, which leads to misleading. We have added caveats in conclusions drawn from Raman and discuss the limitation. Raman may be useful for metals with strong binding with *H, such as Pt and Pd. However, the peak of *H on Cu cannot be observed using Raman due to the extremely weak adsorption of *H on Cu. The advanced characterization methods for directly prove *H coverage are expected to better understand the CO₂RR process. Thus, a direct evidence for *H coverage cannot be supported by Raman, we provided a speculative explanation based on the experimental results.

At present, the indirect methods of proving *H coverage are as follows:

1) *H coverage indirectly proved through water content. Interfacial water content obtained by FT-IR and CLSM (*Nat. Commun.*, 2020, 11, 2038; *J. Am. Chem. Soc.*, 2020, 142, 8748–8754). The speculative reason of this method is that the local concentration of reactants (CO₂, H₂O) can influence the intermediate coverage (*CO, *H), which is supported by previous results (*Nat. Catal.*, 2019, 22, 1124–1131, *Joule* 2020, 4, 1–17; *Nat. Commun.*, 2021, 12, 36). The catalytic reaction is related to both thermodynamics and kinetics. In our work, the catalytic reaction is controlled by kinetics (mass transfer) without changing thermodynamics due to the sluggish mass transfer. Specifically, we strove to increase the reaction rate of one step without strongly modulating the thermodynamic factors.

An independent means of controlling intermediate coverage (*CO, *H) is by changing the local concentration of reactants (CO₂, H₂O). For example, the link between CO coverage (θ_{CO}) and the local CO concentration is given as following Equation 1 (*Nat. Catal.*, 2019, 22, 1124–1131, *Joule* 2020, 4, 1–17; *Nat. Commun.*, 2021, 12, 36):

$$\theta_{CO} = \theta \times [CO] \times e^{-\frac{E_{CO}}{RT}}$$

Where θ is the coverage of free surface sites, E_{CO} is the CO adsorption energy on the surface, R is the ideal gas constant and T is the temperature. The coverage of intermediate (θ_{CO}) is a function of CO adsorption energy (E_{CO}) and local CO concentration ([CO]). At equilibrium, the surface coverage of intermediate (*CO, *H) is directly proportional to the concentration of reactants (CO₂, H₂O). Therefore, it is reasonable to indirectly speculate the *H coverage rate through water content.

2) *H coverage indirectly proved through *CO coverage. *H and *CO occupy most of the Cu surface sites, so they are in direct competition with each other for surface sites. When the surface maintains a high *CO coverage, the corresponding *H coverage will decrease (*J. Am.*

Chem. Soc., 2017, 139,15848-15857; *Angew. Chem. Int. Ed.*, 2018,57, 10221-10225; *J. Phys. Chem. Lett.*, 2016, 7, 1471-1477; *Nat. Commun.*, 2021, 12, 5745). According to the results of in-situ ATR-SEIRAS spectra (Fig. 3d, 3e) and in-situ Raman spectra (Supplementary Fig. 10), higher hydrophobicity indicates higher *CO coverage, therefore the corresponding *H coverage is lower.

In this work, the speculative reasons for *H coverage comes from the two methods mentioned above, namely indirect proof by water content and *CO coverage, and the conclusions (link between wettability and *H coverage) drawn by these two methods are the same. Thanks to the kind reminder from the reviewer, we have realized that we lack an explanation for speculation of *H coverage, and the language when discussing the mechanism is not very reasonable.

Therefore, we have added caveats in conclusions drawn from Raman and discuss the limitation (page 13 and page 22 in revised manuscript; page 36 in Supplementary Information)

Page 13 in MS:

Unfortunately, the peak of *H on Cu cannot be observed by in-situ Raman spectra (Supplementary Fig. 35) or in-situ ATR-SEIRAS spectra (Fig. 3d, 3e) due to the extremely weak adsorption of *H on Cu. At present, *H coverage can be investigated indirectly through *CO coverage or H₂O content^{18,46-49}. *H and *CO occupy most of the Cu surface sites, resulting in direct competition between *H and *CO for surface sites⁴⁷. When the surface maintains a high *CO coverage, the corresponding *H coverage will decrease^{12,18,46,47}. According to the results of in-situ ATR-SEIRAS spectra (Fig. 3d, 3e) and in-situ Raman spectra (Supplementary Fig. 35), higher hydrophobicity indicates higher *CO coverage, thus the corresponding *H coverage is lower. Further, the surface coverage of intermediate (*CO, *H) is directly proportional to the concentration of reactants (CO₂, H₂O). Therefore, the *H coverage can be deduced indirectly through water content.

Page 22 in MS:

In-situ Raman spectroscopy measurements. In-situ Raman spectroscopy was carried out in a custom-designed flow cell, which was manufactured by Gaosunion Co., Ltd., Tianjin. The electrode was encased in a PEEK fitting, with an exposed circular geometric surface area of ~1 cm². A platinum wire and an Ag/AgCl electrode (saturated KCl, Gaosunion Co., Ltd., Tianjin) were used as the counter and the reference electrode, respectively. The counter electrode is separated from the working electrode by an anion exchange membrane (FAA-3-PK-75, Fumatech) to avoid cross-contamination. In situ Raman spectroscopy was performed with a Raman microscopy system (LabRAM HR Evolution, Horiba Jobin Yvon). A 785 nm laser served as the excitation source. Electrochemical measurements were carried out with a potentiostat (CompactStat.e20250, IVIUM).

Page 36 in SI:

Supplementary Fig. 35 | In-situ Raman spectra of Cu and Cu-12C.

Adsorbed CO bands located in the 2000–2150 cm^{-1} region are generally attributed to linearly bound CO. Operando in-situ Raman spectra reveal that there are multiple distinct CO_{ad} sites on the Cu surface, among which one band centered at 2073 cm^{-1} and a weak shoulder at 2089 cm^{-1} are observed. However, the peak of $\ast\text{H}$ on Cu cannot be observed by in-situ Raman spectroscopy due to the extremely weak adsorption of $\ast\text{H}$ on Cu. The advanced characterization methods for directly prove $\ast\text{H}$ coverage are expected to better understand the CO_2RR process.

Moreover, we have added associated speculative reasons (page 13, 16, 17) in the revised manuscript.

Page 16 in MS:

...At equilibrium, the surface coverage of intermediate ($\ast\text{CO}$, $\ast\text{H}$) is directly proportional to the concentration of reactants (CO_2 , H_2O). For example, the relevance between the surface coverage of adsorbed $\ast\text{CO}_x$ ($\theta_{\text{CO}_x, x=1 \text{ or } 2}$) on the catalyst and the local concentration of CO_x ($[\text{CO}_x]$), which is given as Equation 1^{5,42}:

$$\theta_{\text{CO}_x} = \theta \times [\text{CO}_x] \times e^{-\frac{E_{\text{CO}_x}}{RT}} \quad (1)$$

where θ is the coverage of free surface sites, E_{CO_x} is the CO_x adsorption energy on the surface, R is the ideal gas constant and T is the temperature. The coverage of intermediate (θ_{CO_x}) is a function of CO adsorption energy (E_{CO_x}) and local CO concentration ($[\text{CO}_x]$)...

Page 13 in MS:

Unfortunately, the peak of $\ast\text{H}$ on Cu cannot be observed by in-situ Raman spectra (Supplementary Fig. 35) or in-situ ATR-SEIRAS spectra (Fig. 3d, 3e) due to the extremely weak adsorption of $\ast\text{H}$ on Cu. At present, $\ast\text{H}$ coverage can be investigated indirectly through $\ast\text{CO}$

coverage or H₂O content^{18,46-49}. *H and *CO occupy most of the Cu surface sites, resulting in direct competition between *H and *CO for surface sites⁴⁷. When the surface maintains a high *CO coverage, the corresponding *H coverage will decrease^{12,18,46,47}. According to the results of in-situ ATR-SEIRAS spectra (Fig. 3d, 3e) and in-situ Raman spectra (Supplementary Fig. 35), higher hydrophobicity indicates higher *CO coverage, thus the corresponding *H coverage is lower. Further, the surface coverage of intermediate (*CO, *H) is directly proportional to the concentration of reactants (CO₂, H₂O). Therefore, the *H coverage can be deduced indirectly through water content.

Page 17 in MS:

...As analyzed above, the *H coverage can be deduced indirectly through water content...

Further, we have revised the statement (page2, 4, 10 and page14-18) in the revised manuscript.

Page 2, 4, 10, 14-18 in MS:

Based on the aforementioned, the mechanism of interfacial wettability effect on the ethylene and ethanol pathways can be deduced (Fig. 5).

Different interfacial structures influence the mass transport of CO₂ and H₂O, which may lead to the variation of *CO and *H coverage.

...Therefore, it is speculated that the reaction pathways of ethylene and ethanol can be tuned by kinetic-controlled *CO/*H ratio through controllable wettability, which is enabled by modifying alkanethiols with different alkyl chain lengths...

...Therefore, with the increasing of the hydrophobicity, the coverage ratio of *CO/*H may increase simultaneously. The limitation step of CO₂ reduction is also change from insufficient *CO to inadequate *H, which may lead to more ethanol product rather than ethylene.

...Characterization and simulation reveal that the mass transport of CO₂ and H₂O is related with interfacial wettability, which may result in the variation of kinetic-controlled *CO and *H ratio and is relevant with ethylene and ethanol pathways...

...Then, the local concentration of CO₂ and H₂O can be modulated by changing CO₂ and H₂O transport through different interface wettability, which may achieve an optimized equilibrium of kinetic-controlled *CO and *H in a controllable manner...

Our data (Fig. 2) also may imply that the kinetic-controlled *CO/*H ratio can be controlled by tuning the local CO₂/H₂O ratio through changing wettability of the catalyst.

The local CO₂/H₂O ratio derived from wettability may affect the coverage of *CO and *H, resulting in the reaction pathways toward ethylene or ethanol...The exposure of the three-phase interface may balance the gas and liquid mass transport, resulting in an optimized *CO/*H ratio for ethylene and ethanol conversion...A gas-liquid-solid interfaces may balance gas and liquid mass transfer to achieve a suitable *CO/*H ratio for producing ethylene and ethanol.

...Through changing the lengths of alkyl chain on alkanethiols, the equilibrium of kinetic-controlled *CO and *H intermediates on surface **may** be controlled...Interfacial hydrophobic treatment **may** accelerate CO₂ mass transport while hinder H₂O absorption, resulting higher *CO/*H ratio on interface. The reaction limitation **may** shift from *CO insufficiency to *H exhausting, resulting the main product is changed from ethanol to ethylene...

Specific Comments R2-2: *The response(R2-2) is not convincing. This reviewer is not looking for a conclusion based on experimental observation, rather looks for a link between the hypothesis of surface *H/*CO coverage and product distribution. The question is why altering *H/*CO surface coverage, only affects ethylene and ethanol formation, knowing that in experiment we only observe these products. The question is “why” and not “what” we observe.*

Response: We thank the reviewer for insightful comment. We strongly agree with the reviewer that our statement is a hypothesis, thus we have added descriptions to clarify that the discussion is based on speculation. The link for our hypothesis about the surface *H/*CO coverage and product distribution is as following:

First, the surface coverage of intermediate (*CO, *H) is directly proportional to the concentration of reactants (CO₂, H₂O). For example, the relevance between local CO_x (x=1 or 2) concentration ([CO_x]) and the surface coverage of CO_x (θ_{CO_x}) is shown as the following equation (*Nat. Catal.*, 2019, 22, 1124–1131; *Joule*, 2020, 4, 1–17):

$$\theta_{CO_x} = \theta \times [CO_x] \times e^{-\frac{E_{CO_x}}{RT}} \quad (1)$$

where θ is the coverage of free surface sites, E_{CO_x} is the CO_x adsorption energy on the surface, R is the ideal gas constant and T is the temperature. The coverage of intermediate (θ_{CO_x}) is a function of CO adsorption energy (E_{CO_x}) and local CO concentration ([CO_x]). A promising means of controlling CO_x coverage is to control the CO partial pressure (p_{CO_x}) or local CO_x concentration ([CO_x]). In this work, we strove to increase the reaction rate of one step without strongly modulating the others, by judiciously increasing the coverage of one key intermediate, yet not interfering with the electronic structure (hence binding strength) of Cu. In brief, the catalytic reaction is controlled by kinetics (mass transport) without changing thermodynamics. As in previous report, some molecular-metal composite was developed to generates a high concentration of the key early intermediate, yet does not modulate the metallic active sites germane to the crucial C–C coupling step, which is essentially similar to our work, but through a different method (*Nat. Catal.*, 2020, 3, 75–82).

Further, local CO₂ concentration can influence the surface coverage of *CO and *H, which affects the reaction pathways toward multi-carbon (C₂₊) products (*Nat. Catal.*, 2019, 22, 1124–1131; *Joule*, 2020, 4, 1–17). *CO coverage has been widely accepted as one crucial factor on selective produce ethylene (*Joule*, 2020, 4, 1104-1120; *Nat. Nanotechnol.*, 2019, 14, 1063-1070). Tuning the *H coverage is an effective approach to realized high-efficiency CO₂-to-ethanol conversion (*Nat. Commun.*, 2019, 10, 5814; *Joule*, 2020, 4, 1688-1699) which is less relevance with *CO coverage (*Joule*, 2020, 4, 1104-1120). Therefore, the *CO and *H intermediates on catalyst surface can affect the pathways of ethylene and ethanol. Inspired by previous works, we speculate that the variation of *H and *CO coverage caused by the local concentration of CO₂ and H₂O via wettability may be one of the important reasons for the selectivity changes in ethylene and ethanol.

As the reviewer speculated, altering *H/*CO surface coverage not only affects ethylene and ethanol formation, but also affects other products. Because of the properties of Cu based catalysts, ethylene and ethanol are the main products. Compared with ethylene and ethanol, the selectivity of other products is relatively less affected by the *CO and *H coverage due to the low selectivity. In this work, we mainly discuss the main products, namely ethylene and ethanol.

We apologize that we did not explain speculative reasons in the manuscript. To avoid misleading, we have added speculative reasons (page 16, 17) in the revised manuscript.

Page 16 in MS:

...At equilibrium, the surface coverage of intermediate (*CO, *H) is directly proportional to the concentration of reactants (CO₂, H₂O). For example, the relevance between the surface coverage of adsorbed *CO_x ($\theta_{CO_x, x=1 \text{ or } 2}$) on the catalyst and the local concentration of CO_x ([CO_x]), which is given as Equation 1^{5,42}:

$$\theta_{CO_x} = \theta \times [CO_x] \times e^{-\frac{E_{CO_x}}{RT}} \quad (1)$$

where θ is the coverage of free surface sites, E_{CO_x} is the CO_x adsorption energy on the surface, R is the ideal gas constant and T is the temperature. The coverage of intermediate (θ_{CO_x}) is a function of CO adsorption energy (E_{CO_x}) and local CO concentration ([CO_x])...

Page 16 in MS:

...*CO coverage and *H coverage has been widely accepted as one crucial factor on selective production of ethylene or ethanol⁵⁻⁸. Namely, the *CO and *H intermediates on catalyst surface can affect the pathways of ethylene and ethanol. Inspired by previous works⁵⁻⁸, we speculate that the variation of *H and *CO coverage caused by the local concentration of CO₂ and H₂O via wettability may be one of the important reasons for the selectivity changes in ethylene and ethanol.

Page 17 in MS:

...As analyzed above, the *H coverage can be deduced indirectly through water content...

Moreover, we have emphasized the impact of *CO and *H coverage on other products on page 16 in the revised manuscript.

Page 16 in MS:

...It is worth noting that altering *CO/*H surface coverage not only affects ethylene and ethanol formation, but also affects other products. Herein, only the main products, namely ethylene and ethanol, are discussed...

Further, we have cleared in the manuscript that the discussion is based on speculation (page2, 4, 10 and page14-18).

Page 2, 4, 10, 14-18 in MS:

Based on the aforementioned, the mechanism of interfacial wettability effect on the ethylene and ethanol pathways can be deduced (Fig. 5).

Different interfacial structures influence the mass transport of CO₂ and H₂O, which may lead to the variation of *CO and *H coverage.

...Therefore, it is speculated that the reaction pathways of ethylene and ethanol can be tuned by kinetic-controlled *CO/*H ratio through controllable wettability, which is enabled by modifying alkanethiols with different alkyl chain lengths...

...Therefore, with the increasing of the hydrophobicity, the coverage ratio of *CO/*H may increase simultaneously. The limitation step of CO₂ reduction is also change from insufficient *CO to inadequate *H, which may lead to more ethanol product rather than ethylene.

...Characterization and simulation reveal that the mass transport of CO₂ and H₂O is related with interfacial wettability, which may result in the variation of kinetic-controlled *CO and *H ratio and is relevant with ethylene and ethanol pathways...

...Then, the local concentration of CO₂ and H₂O can be modulated by changing CO₂ and H₂O transport through different interface wettability, which may achieve an optimized equilibrium of kinetic-controlled *CO and *H in a controllable manner...

Our data (Fig. 2) also may imply that the kinetic-controlled *CO/*H ratio can be controlled by tuning the local CO₂/H₂O ratio through changing wettability of the catalyst.

The local CO₂/H₂O ratio derived from wettability may affect the coverage of *CO and *H, resulting in the reaction pathways toward ethylene or ethanol...The exposure of the three-phase interface may balance the gas and liquid mass transport, resulting in an optimized *CO/*H ratio for ethylene and ethanol conversion...A gas-liquid-solid interfaces may balance gas and liquid mass transfer to achieve a suitable *CO/*H ratio for producing ethylene and ethanol.

...Through changing the lengths of alkyl chain on alkanethiols, the equilibrium of kinetic-controlled *CO and *H intermediates on surface may be controlled...Interfacial hydrophobic treatment may accelerate CO₂ mass transport while hinder H₂O absorption, resulting higher *CO/*H ratio on interface. The reaction limitation may shift from *CO insufficiency to *H exhausting, resulting the main product is changed from ethanol to ethylene...